# PTQ4DiT: Post-training Quantization for Diffusion Transformers

**Junyi Wu**[1,3,*] **Haoxuan Wang**[1,*] **Yuzhang Shang**[2] **Mubarak Shah**[3] **Yan Yan**[1,†]

[1]University of Illinois Chicago   [2]Illinois Institute of Technology   [3]University of Central Florida

https://github.com/adreamwu/PTQ4DiT

## Abstract

The recent introduction of Diffusion Transformers (DiTs) has demonstrated exceptional capabilities in image generation by using a different backbone architecture, departing from traditional U-Nets and embracing the scalable nature of transformers. Despite their advanced capabilities, the wide deployment of DiTs, particularly for real-time applications, is currently hampered by considerable computational demands at the inference stage. Post-training Quantization (PTQ) has emerged as a fast and data-efficient solution that can significantly reduce computation and memory footprint by using low-bit weights and activations. However, its applicability to DiTs has not yet been explored and faces non-trivial difficulties due to the unique design of DiTs. In this paper, we propose **PTQ4DiT**, a specifically designed PTQ method for DiTs. We discover two primary quantization challenges inherent in DiTs, notably the presence of salient channels with extreme magnitudes and the temporal variability in distributions of salient activation over multiple timesteps. To tackle these challenges, we propose **C**hannel-wise **S**alience **B**alancing (**CSB**) and **S**pearman's $\rho$-guided **S**alience **C**alibration (**SSC**). CSB leverages the complementarity property of channel magnitudes to redistribute the extremes, alleviating quantization errors for both activations and weights. SSC extends this approach by dynamically adjusting the balanced salience to capture the temporal variations in activation. Additionally, to eliminate extra computational costs caused by PTQ4DiT during inference, we design an offline re-parameterization strategy for DiTs. Experiments demonstrate that our PTQ4DiT successfully quantizes DiTs to 8-bit precision (W8A8) while preserving comparable generation ability and further enables effective quantization to 4-bit weight precision (W4A8) for the first time.

## 1 Introduction

Diffusion models have spearheaded recent breakthroughs in generation tasks [59, 7]. In the past, these models were based on convolutional U-Nets [40] as their backbone architectures [46, 17, 9, 39]. However, recent work [2, 60, 30] has revealed that the U-Net inductive bias is not essential for the success of diffusion models and even limits their scalability. Among this trend, Diffusion Transformers (DiTs) [37] have demonstrated exceptional capabilities in image generation by using a different backbone architecture. Different from U-Nets that carefully design downsampling and upsampling blocks with skip-connections, DiTs are constructed by repeatedly and sequentially stacking transformer blocks [49]. This architectural choice inherits the scaling property of transformers [5, 48, 58, 31], facilitating more flexible parameter expansion for enhanced performance. With their versatility and scalability, DiTs have been successfully integrated into advanced frameworks like Sora [4], demonstrating their potential as a leading architecture for future generative models [14, 6, 30, 65].

---

*Equal Contribution. †Corresponding Author. Work done during Junyi Wu's visit to CRCV, UCF.

38th Conference on Neural Information Processing Systems (NeurIPS 2024).

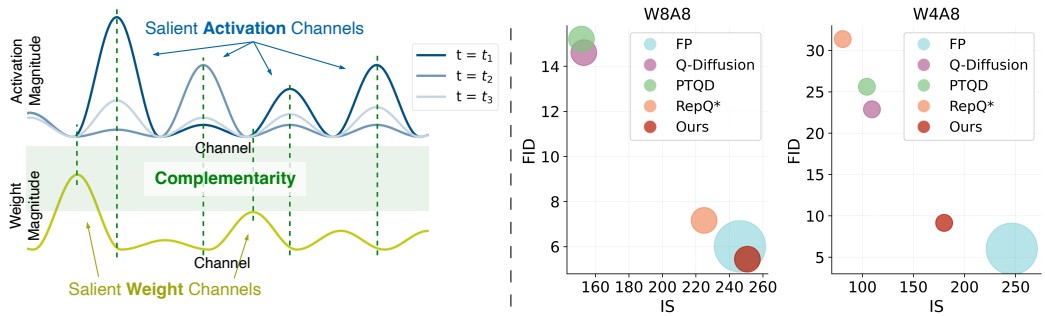

Figure 1: **(Left)** Illustration of salient channels in **activation** and **weight**. Note that salient activation channels exhibit variations over different timesteps (*e.g.*, $t = t_1, t_2, t_3$.), posing non-trivial quantization challenges. To mitigate the overall quantization difficulty, our method leverages the **complementarity** (activation and weight channels do not have extreme magnitude simultaneously) to redistribute channel salience between weights and activations across various timesteps. **(Right)** Quantization performance on W8A8 and W4A8, employing FID (lower is better) and IS (higher is better) metrics on ImageNet $256 \times 256$ [41]. The circle size indicates the model size.

Nonetheless, the widespread adoption of Diffusion Transformers is currently constrained by their massive amount of parameters and computational complexity. DiTs consist of a large number of repeated transformer blocks and employ a lengthy iterative image sampling process, demanding high computational costs during inference. For instance, generating a $512 \times 512$ resolution image using DiTs can take more than 20 seconds and $10^5$ Gflops on an NVIDIA RTX A6000 GPU. This substantial requirement makes them unacceptable or impractical for real-time applications, especially considering the potential for increased model sizes and feature resolutions.

Model quantization [33, 32, 28] is a prominent technique for accelerating deep learning models because of its high compression rate and significant reduction in inference time. This technique transforms model weights and activations into low-bit formats, which directly reduces the computational burden and memory usage. Among various methods, Post-training Quantization (PTQ) stands out as a leading approach since it circumvents the need to re-train the original model [62, 44, 18, 63, 22]. Practically, PTQ requires only a small dataset for fast calibration, thus is highly suitable for quantizing DiTs, whose re-training process involves extensive data and computational resources [14, 6].

However, quantizing DiTs in a post-training manner is non-trivial due to the complex distribution patterns in weights and activations. We discover two major challenges that impede the effective quantization of DiTs: ❶ The emergence of *salient channels*, channels with extreme magnitudes, in both weights and activations of linear layers within DiT blocks. When low-bit representations are used for these salient channels, pronounced errors compared to the full-precision (FP) counterparts are observed, incurring fundamental difficulty for quantization. ❷ The extreme magnitudes within salient activation channels significantly vary as the inference proceeds across multiple timesteps. This dynamic behavior further complicates the quantization of salient channels, as quantization strategies optimized for one timestep may fail to generalize to other timesteps. Such inconsistency, especially in salient channels that dominate the activation signals, can result in significant deviations from the full-precision distribution, leading to degradation in the generation ability of quantized models.

Targeting these two challenges, we propose a novel Post-training Quantization method specifically for Diffusion Transformers, termed **PTQ4DiT**. To address the quantization difficulty associated with salient channels, we propose **C**hannel-wise **S**alience **B**alancing (**CSB**). CSB capitalizes on an interesting observation of the salient channels that extreme values do not coincide in the same channel of activation and weight within the same layer, as shown in Figure 1 (Left). Leveraging this complementarity property, CSB facilitates the redistribution of extreme magnitudes between activations and weights to minimize the overall channel salience. Concretely, we introduce Salience Balancing Matrices, derived from the statistical properties of activation and weight distributions, to channel-wise transform both activations and weights. This transformation achieves equilibrium in their salient channels, effectively mitigating the quantization difficulty of the balanced distributions.

Recognizing the variability in activations over different timesteps, we further extend the concept of channel salience along the temporal dimension and propose **S**pearman's $\rho$-guided **S**alience **C**alibration (**SSC**). This method refines the Salience Balancing Matrices to comprehensively evaluate activation

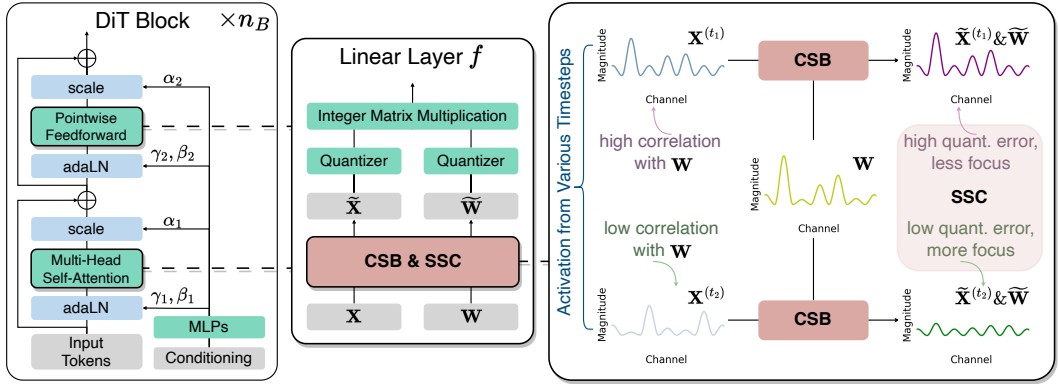

Figure 2: **(Left)** Overview of the Diffusion Transformer (DiT) Block [37]. **(Middle)** Illustration of the linear layer in Multi-Head Self-Attention (MHSA) and Pointwise Feedforward (PF) modules, which incorporates our proposed Channel-wise Salience Balancing (CSB) and Spearman's $\rho$-guided Salience Calibration (SSC) to address quantization difficulties for both activation $\mathbf{X}$ and weight $\mathbf{W}$. Appendix A depicts detailed structures of the MHSA and PF modules with adjusted linear layers. **(Right)** Illustration of CSB and SSC in PTQ4DiT. CSB redistributes salient channels between weights and activations from various timesteps to reduce overall quantization errors. SSC calibrates the activation salience across multiple timesteps via selective aggregation, with more focus on timesteps where quantization errors can be significantly reduced by CSB.

salience over timesteps, with more emphasis on timesteps where the complementarity between salient activation and weight channels is more significant. Furthermore, we design a re-parameterization scheme that can offline absorb these Salience Balancing Matrices into adjacent layers, thus avoiding additional computation overhead at the inference stage.

While the performance of mainstream PTQ methods degrades on DiTs, our PTQ4DiT achieves comparable performance to the FP counterpart with 8-bit weight and activation (W8A8). In addition, PTQ4DiT can generate high-quality images with further reduced weight precision at 4-bit (W4A8). To the best of our knowledge, PTQ4DiT is the first method for effective DiT quantization.

## 2 Backgrounds and Related Works

### 2.1 Diffusion Transformers

Although generative models built upon U-Nets have made great advancements in the last few years, transformer-like architectures are increasingly attracting attention [39, 7, 59]. The recently explored Diffusion Transformers (DiTs) [37] have achieved state-of-the-art performance in image generation. Encouragingly, DiTs exhibit remarkable scalability in model size and data representation, positioning them as a promising backbone for a wide range of generative applications [4, 30, 65].

DiTs consist of $n_B$ blocks, each containing a Multi-Head Self-Attention (MHSA) and a Pointwise Feedforward (PF) module [49, 11, 37], both preceded by their respective adaptive Layer Norm (adaLN) [38]. We illustrate the DiT Block structure in Figure 2 (Left). These blocks sequentially process the noised latent and conditional information, which are both represented as tokens in a lower-dimensional latent space [39]. In each block, conditional input $\mathbf{c} \in \mathbb{R}^{d_{in}}$ is converted to scale and shift parameters ($\boldsymbol{\gamma}, \boldsymbol{\beta} \in \mathbb{R}^{d_{in}}$), which are regressed through MLPs then injected into the noised latent $\mathbf{Z} \in \mathbb{R}^{n \times d_{in}}$ via adaLN:

$$(\boldsymbol{\gamma}, \boldsymbol{\beta}) = \text{MLPs}(\mathbf{c}), \quad \text{adaLN}(\mathbf{Z}) = \text{LN}(\mathbf{Z}) \odot (\mathbf{1} + \boldsymbol{\gamma}) + \boldsymbol{\beta}, \tag{1}$$

where $\text{LN}(\cdot)$ is the standard Layer Norm [1]. These adaLN modules dynamically adjust the layer normalization before each MHSA and PF module, enhancing DiTs' adaptability to varying conditions and improving the generation quality.

Despite their effectiveness, DiTs require extensive computational resources to generate high-quality images, which impedes their real-world deployment. In this paper, we devise a model quantization method for DiTs that reduces both time and memory consumption without necessitating re-training the original models, offering a robust and practical solution for enhancing the efficiency of DiTs.

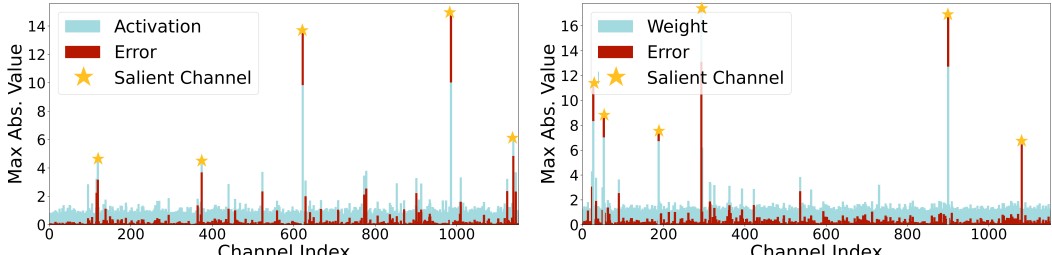

Figure 3: Illustration of maximal absolute magnitudes of activation (**left**) and weight (**right**) channels in a DiT linear layer, alongside their corresponding quantization Error (MSE). Channels with greater maximal absolute values tend to incur larger errors, presenting a fundamental quantization difficulty.

## 2.2 Model Quantization

Model quantization is a compression technique that improves the inference efficiency of deep learning models by transforming full-precision tensors into $b$-bit integer approximations, leading to direct computational acceleration and memory saving [33, 62, 8, 28, 19, 64]. Formally, the quantization process can be defined as:

$$Q(\mathbf{x}) = \text{clamp}(\lfloor \frac{\mathbf{x}}{\boldsymbol{\delta}} \rceil + \boldsymbol{\lambda}, 0, 2^b - 1), \quad (2)$$

where $\mathbf{x}$ denotes the full-precision tensor, $\lfloor \cdot \rceil$ is the round-to-nearest operator [32], and the clamp function restricts the quantized value within the range of $[0, 2^b - 1]$. Here, $\boldsymbol{\delta}$ and $\boldsymbol{\lambda}$ are quantization parameters subject to optimization. Among various quantization methods, Post-training Quantization (PTQ) is a dominant approach for large quantized models, as it circumvents the substantial resources required for model re-training [20, 52, 25, 44, 15]. PTQ employs a small calibration dataset to optimize quantization parameters, which aims to reduce the performance gap between the quantized models and their full-precision counterparts with minimal data and computational expenses.

PTQ has been effectively applied to a wide range of neural networks, including CNNs [20, 52, 25], Language Transformers [8, 57, 24, 23, 27], Vision Transformers [62, 13, 21, 29], and U-Net-based Diffusion models [44, 18, 51, 50]. Despite its demonstrated success, PTQ's applicability to Diffusion Transformers (DiTs) remains unexplored, presenting a significant open challenge within the research community. To bridge this gap, our work delves into the unique challenges of quantizing DiTs and introduces the first PTQ method for DiTs that can fruitfully preserve their generation performance.

## 3 Diffusion Transformer Quantization Challenges

Diffusion Transformers (DiTs) diverge from conventional generative or discriminative models [39, 11] through their unique design. Specifically, DiTs are constructed with a series of large transformer blocks and operate under a multi-timestep paradigm to progressively transform pure noise into images. Our analysis reveals complex distribution patterns and temporal dynamics in the inference process of DiTs, identifying two primary challenges that prevent effective DiT quantization.

❶ **Pronounced Quantization Error in Salient Channels.** The first challenge lies in systematic quantization errors in DiT's linear layers. As shown in Figure 3, activation and weight channels with significantly high absolute values are prone to substantial errors after quantization. We term these as *salient channels*, characterized by extreme values that greatly exceed the typical range of magnitudes. Upon uniform quantization (Eq. (2)), it is often necessary to truncate these extreme values in order to maintain the precision of the broader set of standard channels. This compromise can result in notable deviations from the original full-precision distribution as the sampling process proceeds, especially given DiT's layered architecture and repetitive inference paradigm.

❷ **Temporal Variation in Salient Activation.** Another challenge of DiT quantization arises from temporal variations in the magnitudes of salient activation channels. Rather than static inputs, DiTs operate across a sequence of timesteps to generate high-quality images from random noise. Consequently, activation distributions can vary drastically within the inference process, which is particularly evident in salient channels that dominate the signal. Figure 4 demonstrates that the distribution of maximal absolute values in activation channels exhibits significant variations over different timesteps. This temporal variability introduces a non-trivial difficulty to quantization optimization: Quantization parameters effective for salient activation channels at one timestep may

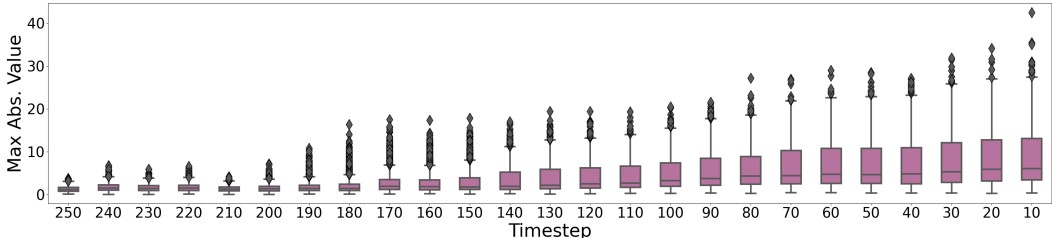

Figure 4: Boxplot of maximal absolute magnitudes of activation channels in a linear layer within DiT over different timesteps, which exhibit significant temporal variations.

not be suitable at other timesteps. Such discrepancies can exacerbate quantization errors, cumulatively impairing the generation quality. Therefore, for accurate quantization, it is imperative to capture the evolving trait of salient channels throughout the entire denoising procedure.

## 4 PTQ4DiT

To overcome the identified challenges, we propose Channel-wise Salience Balancing (CSB) and Spearman's $\rho$-guided Salience Calibration (SSC) in our PTQ4DiT in Sections 4.1 and 4.2, respectively. Subsequently, we devise a re-parameterization scheme in Section 4.3, eliminating extra computational demands of PTQ4DiT during inference while maintaining the mathematical equivalence.

### 4.1 Channel-wise Salience Balancing

A linear layer $f(\cdot; \mathbf{W})$ within MHSA and PF modules typically takes a token sequence $\mathbf{X} \in \mathbb{R}^{n \times d_{in}}$ as input and performs linear transformation with its weight matrix $\mathbf{W} \in \mathbb{R}^{d_{in} \times d_{out}}$, formulated as $f(\mathbf{X}; \mathbf{W}) = \mathbf{X} \cdot \mathbf{W}$, where $n$ is the sequence length, and $d_{in}$ and $d_{out}$ denote the input and output dimensions, respectively. As discussed in Section 3, both the activation $\mathbf{X}$ and the weight matrix $\mathbf{W}$ exhibit salient channels that possess elements with significantly greater absolute magnitudes, which lead to large post-quantization errors.

Fortunately, large values do not coincide in the same channels of activation and weight, so these extremes do not amplify each other, as observed in Figure 3. This property suggests the feasibility of *complementarily* redistributing the large magnitudes in salient channels between activation and weight, thereby alleviating quantization difficulties for both. Inspired by previous works on large model compression [54, 45, 61, 23], we propose **C**hannel-wise **S**alience **B**alancing (**CSB**), which employs diagonal Salience Balancing Matrices $\mathbf{B}^{\mathbf{X}}$ and $\mathbf{B}^{\mathbf{W}}$ to adjust the channel-wise distribution of activation and weight, as expressed by:

$$\widetilde{\mathbf{X}} = \mathbf{X}\mathbf{B}^{\mathbf{X}}, \quad \widetilde{\mathbf{W}} = \mathbf{B}^{\mathbf{W}}\mathbf{W}. \tag{3}$$

To address the quantization difficulties, we need to achieve balanced distributions in $\widetilde{\mathbf{X}}$ and $\widetilde{\mathbf{W}}$, which requires $\mathbf{B}^{\mathbf{X}}$ and $\mathbf{B}^{\mathbf{W}}$ to capture the characteristics of salient channels. Considering that the quantization error is significantly influenced by the range of distributions [33, 57, 26], we measure the *salience s* of an activation or weight channel as the maximal absolute value among its elements:

$$s(\mathbf{X}_j) = \max(|\mathbf{X}_j|), \quad s(\mathbf{W}_j) = \max(|\mathbf{W}_j|), \quad \text{where} \quad j = 1, 2, \ldots, d_{in}. \tag{4}$$

Here, $j$ is the channel index. Consequently, the *balanced salience* $\widetilde{s}$, representing the equilibrium between activation and weight channels, can be quantified using the geometric mean. Specifically, for the $j$-th channel, the balanced salience is calculated as follows:

$$\widetilde{s}(\mathbf{X}_j, \mathbf{W}_j) = (s(\mathbf{X}_j) \cdot s(\mathbf{W}_j))^{\frac{1}{2}}. \tag{5}$$

Building on these concepts, we proceed to construct the Salience Balancing Matrices, which modulate the salience of activations and weights with the guidance of $\widetilde{s}$:

$$\mathbf{B}^{\mathbf{X}} = \text{diag}(\frac{\widetilde{s}(\mathbf{X}_1, \mathbf{W}_1)}{s(\mathbf{X}_1)}, \frac{\widetilde{s}(\mathbf{X}_2, \mathbf{W}_2)}{s(\mathbf{X}_2)}, \ldots, \frac{\widetilde{s}(\mathbf{X}_{d_{in}}, \mathbf{W}_{d_{in}})}{s(\mathbf{X}_{d_{in}})}), \tag{6}$$

$$\mathbf{B}^{\mathbf{W}} = \text{diag}(\frac{\widetilde{s}(\mathbf{X}_1, \mathbf{W}_1)}{s(\mathbf{W}_1)}, \frac{\widetilde{s}(\mathbf{X}_2, \mathbf{W}_2)}{s(\mathbf{W}_2)}, \ldots, \frac{\widetilde{s}(\mathbf{X}_{d_{in}}, \mathbf{W}_{d_{in}})}{s(\mathbf{W}_{d_{in}})}). \tag{7}$$

Following these, the balancing transformation defined by Eq. (3) will result in a complementary redistribution of channel salience between activations and weights. Specifically, for each channel $j$, we have $s(\widetilde{\mathbf{X}}_j) = s(\widetilde{\mathbf{W}}_j) = \widetilde{s}(\mathbf{X}_j, \mathbf{W}_j)$, thereby alleviating the quantization difficulties, as demonstrated by the reduction in overall channel salience:

$$\max(s_o(\widetilde{\mathbf{X}}), s_o(\widetilde{\mathbf{W}})) \leq \max(s_o(\mathbf{X}), s_o(\mathbf{W})). \tag{8}$$

Here, we characterize the overall salience $s_o$ of activations or weights using the maximum salience across channels, *e.g.*, $s_o(\mathbf{X}) = \max(s(\mathbf{X}_1), s(\mathbf{X}_2), \dots, s(\mathbf{X}_{d_{in}}))$, which reflects the distribution range of elements that are quantized collectively under certain granularity.

## 4.2 Spearman's $\rho$-guided Salience Calibration

Diffusion Transformers (DiTs) utilize an iterative denoising process for image sampling [37]. Under this sequential paradigm, the linear layer $f$ receives inputs from an activation sequence $\mathbf{X}^{(1:T)} = (\mathbf{X}^{(1)}, \mathbf{X}^{(2)}, \dots, \mathbf{X}^{(T)})$, which encompasses $T$ timesteps. Targeting a certain timestep $t$, the salience of all activation and weight channels can be evaluated using Eq. (4):

$$\mathbf{s}(\mathbf{X}^{(t)}) = (s(\mathbf{X}_1^{(t)}), s(\mathbf{X}_2^{(t)}), \dots, s(\mathbf{X}_{d_{in}}^{(t)})), \quad \mathbf{s}(\mathbf{W}) = (s(\mathbf{W}_1), s(\mathbf{W}_2), \dots, s(\mathbf{W}_{d_{in}})). \tag{9}$$

While $\mathbf{s}(\mathbf{W})$ remains consistent, we find that $\{\mathbf{s}(\mathbf{X}^{(t)})\}_{t=1}^T$ exhibits significant temporal variations during the process of transforming purely random noise into high-quality images, as demonstrated in Figure 4. These fluctuations diminish the effectiveness of our CSB since quantization errors can be exacerbated by the biased estimation of activation salience among timesteps, resulting in degraded generation quality of the quantized models.

To accurately gauge the activation channel salience under multi-timestep scenarios, we propose **S**pearman's $\rho$-guided **S**alience **C**alibration (SSC). This offers a comprehensive evaluation of activation salience, with enhanced focus allocated to the timesteps where the complementarity property is more significant, facilitating effective salience balancing between activation and weight channels. Essentially, the lower the correlation between activation salience $\mathbf{s}(\mathbf{X}^{(t)})$ and weight salience $\mathbf{s}(\mathbf{W})$, the greater reduction effect in overall channel salience (Eq. (8)). The intuition of SSC is visualized in Figure 2 (Right). Mathematically, we formulate the *Spearman's $\rho$-calibrated Temporal Salience* $\mathbf{s}_\rho$ by selectively aggregating the activation salience along timesteps:

$$\mathbf{s}_\rho(\mathbf{X}^{(1:T)}) = (\eta_1, \eta_2, \dots, \eta_T) \cdot (\mathbf{s}(\mathbf{X}^{(1)}), \mathbf{s}(\mathbf{X}^{(2)}), \dots, \mathbf{s}(\mathbf{X}^{(T)}))^{\mathsf{T}} \in \mathbb{R}^{d_{in}}, \tag{10}$$

where weighting factors $\{\eta_t\}_{t=1}^T$ are derived from a normalized exponential form of inverse Spearman's $\rho$ statistic [47, 55, 56]:

$$\eta_t = \frac{\exp[-\rho(\mathbf{s}(\mathbf{X}^{(t)}), \mathbf{s}(\mathbf{W}))]}{\sum_{\tau=1}^T \exp[-\rho(\mathbf{s}(\mathbf{X}^{(\tau)}), \mathbf{s}(\mathbf{W}))]}. \tag{11}$$

Here, $\rho(\cdot, \cdot)$ computes the correlation between two sequences, and $\eta_t$ serves as the weighting factor for activation salience at timestep $t$. In this method, $\eta_t$ inversely reflects the correlation coefficient $\rho(\mathbf{s}(\mathbf{X}^{(t)}), \mathbf{s}(\mathbf{W}))$, thereby prioritizing timesteps where there is a higher degree of complementarity in salience between activations and weights. Subsequently, we utilize $\mathbf{s}_\rho$ for activation salience in Eqs. (5), (6), and (7), yielding refined Salience Balancing Matrices, denoted as $\mathbf{B}_\rho^{\mathbf{X}}$ and $\mathbf{B}_\rho^{\mathbf{W}}$. By applying SSC, we calibrate the activation salience within CSB to strategically account for the temporal variations during the denoising process. Appendix B presents the full Algorithm for PTQ4DiT.

## 4.3 Re-Parameterization

Before quantization, we estimate $\mathbf{B}_\rho^{\mathbf{X}}$ and $\mathbf{B}_\rho^{\mathbf{W}}$ on a small calibration dataset generated from multiple timesteps. Then, we incorporate these matrices into the linear layers within MHSA and PF modules [37] to alleviate the quantization difficulty. Given that $\mathbf{B}_\rho^{\mathbf{X}}$ and $\mathbf{B}_\rho^{\mathbf{W}}$ are mutual inverses, this incorporation maintains mathematical equivalence to the original linear layer $f$:

$$\widetilde{\mathbf{X}} \cdot \widetilde{\mathbf{W}} = (\mathbf{X}\mathbf{B}_\rho^{\mathbf{X}}) \cdot (\mathbf{B}_\rho^{\mathbf{W}}\mathbf{W}) = \mathbf{X} \cdot \mathbf{W}. \tag{12}$$

The proof is provided in Appendix C. Furthermore, we design a re-parameterization scheme for DiTs, allowing for obtaining $\widetilde{\mathbf{X}}$ and $\widetilde{\mathbf{W}}$ without extra computational burden during inference. Specifically,

| RepQ* (W4A8) | Q-Diffusion (W4A8) | **PTQ4DiT (W4A8)** |

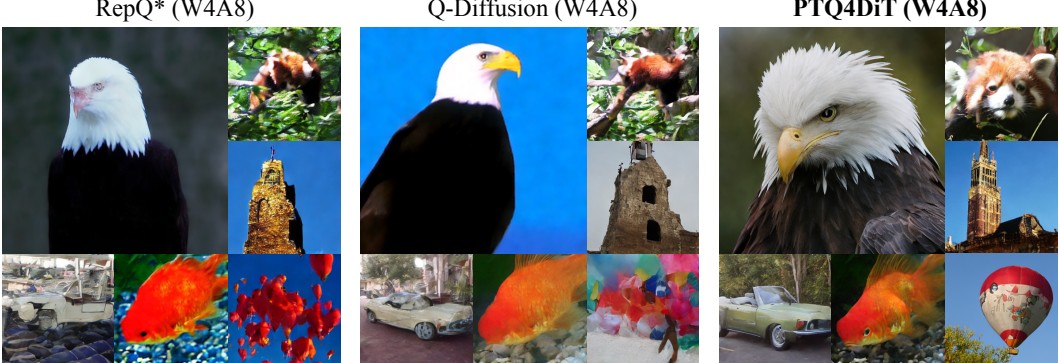

Figure 5: Random samples generated by PTQ4DiT and two strong baselines: RepQ* [21] and Q-Diffusion [18], with W4A8 quantization on ImageNet 512×512 and 256×256. Our method can produce high-quality images with finer details. Appendix E presents more visualization results.

we update the weight matrix of linear layer $f$ to $\widetilde{\mathbf{W}}$ offline and seamlessly integrate $\mathbf{B}_\rho^{\mathbf{X}}$ into the preceding linear transformation operations. This integration includes adaptations to adaLN [38, 37] and matrix multiplications within attention mechanisms [49]. Appendix A discusses these adaptations.

**Post-adaLN.** For linear layers following the adaLN module, we integrate $\mathbf{B}_\rho^{\mathbf{X}}$ by adjusting the scale and shift parameters ($\boldsymbol{\gamma}, \boldsymbol{\beta} \in \mathbb{R}^{d_{in}}$) within adaLN:

$$\widetilde{\mathbf{X}} = \widetilde{\mathrm{adaLN}}(\mathbf{Z}) = \mathrm{LN}(\mathbf{Z}) \odot (\mathbf{B}_\rho^{\mathbf{X}} + \widetilde{\boldsymbol{\gamma}}) + \widetilde{\boldsymbol{\beta}}, \quad \text{where} \quad \widetilde{\boldsymbol{\gamma}} = \boldsymbol{\gamma}\mathbf{B}_\rho^{\mathbf{X}}, \quad \widetilde{\boldsymbol{\beta}} = \boldsymbol{\beta}\mathbf{B}_\rho^{\mathbf{X}}. \quad (13)$$

Equivalently, we fuse $\mathbf{B}_\rho^{\mathbf{X}}$ into the MLPs responsible for regressing these parameters, thus avoiding additional computation overhead at inference time. Detailed derivations are provided in Appendix D.

**Post-Matrix-Multiplication.** For linear layers after matrix multiplication, the effect of PTQ4DiT can be realized by directly absorbing the Salience Balancing Matrices into the preceding de-quantization functions associated with the matrix multiplication [12, 53, 61].

## 5 Experiments

### 5.1 Experimental Settings

Our experimental setup is similar to the original study of Diffusion Transformers (DiTs) [37]. We evaluate PTQ4DiT on the ImageNet dataset [41], using pre-trained class-conditional DiT-XL/2 models [37] at image resolutions of 256×256 and 512×512. The DDPM solver [17] with 250 sampling steps is employed for the generation process. To further assess the robustness of our method, we conduct additional experiments with reduced sampling steps of 100 and 50.

For fair benchmarking, all methods utilize uniform quantizers for all activations and weights, with channel-wise quantization for weights and tensor-wise for activations, unless specified otherwise. To construct the calibration set, we uniformly select 25 timesteps for 256-resolution experiments and 10 timesteps for 512-resolution experiments, generating 32 samples at each selected timestep. The optimization of quantization parameters follows the implementation from Q-Diffusion [18]. Our code is based on PyTorch [36], and all experiments are conducted on NVIDIA RTX A6000 GPUs.

To comprehensively assess generated image quality, we employ four metrics: Fréchet Inception Distance (FID) [16], spatial FID (sFID) [42, 34], Inception Score (IS) [42, 3], and Precision, all computed using the ADM toolkit [10]. For all methods under evaluation, including the full-precision (FP) models, we sample 10,000 images for ImageNet 256×256, and 5,000 for ImageNet 512×512, consistent with conventions from prior studies [35, 44].

### 5.2 Quantization Performance

We present a comprehensive assessment of our PTQ4DiT against prevalent baseline methods in various settings. Our evaluation focuses on mainstream Post-training Quantization (PTQ) methods that are widely used and adaptable to DiTs, including PTQ4DM [44], Q-Diffusion [18], and PTQD [15].

Table 1: Performance comparison on ImageNet 256×256. '(W/A)' indicates that the precision of weights and activations are W and A bits, respectively.

| Timesteps | Bit-width (W/A) | Method | Size (MB) | FID ↓ | sFID ↓ | IS ↑ | Precision ↑ |
|---|---|---|---|---|---|---|---|
| | 32/32 | FP | 2575.42 | 4.53 | 17.93 | 278.50 | 0.8231 |
| | | PTQ4DM | 645.72 | 21.65 | 100.14 | 134.22 | 0.6342 |
| | | Q-Diffusion | 645.72 | 5.57 | 18.22 | 227.50 | 0.7612 |
| | 8/8 | PTQD | 645.72 | 5.69 | 18.42 | 224.26 | 0.7594 |
| | | RepQ* | 645.72 | **4.51** | 18.01 | 264.68 | 0.8076 |
| 250 | | **Ours** | 645.72 | 4.63 | **17.72** | **274.86** | **0.8299** |
| | | PTQ4DM | 323.79 | 72.58 | 52.39 | 35.79 | 0.2642 |
| | | Q-Diffusion | 323.79 | 15.31 | 26.04 | 134.71 | 0.6194 |
| | 4/8 | PTQD | 323.79 | 16.45 | **22.29** | 130.45 | 0.6111 |
| | | RepQ* | 323.79 | 23.21 | 28.58 | 104.28 | 0.4640 |
| | | **Ours** | 323.79 | **7.09** | 23.23 | **201.91** | **0.7217** |
| | 32/32 | FP | 2575.42 | 5.00 | 19.02 | 274.78 | 0.8149 |
| | | PTQ4DM | 645.72 | 15.36 | 79.31 | 172.37 | 0.6926 |
| | | Q-Diffusion | 645.72 | 7.93 | 19.46 | 202.84 | 0.7299 |
| | 8/8 | PTQD | 645.72 | 8.12 | 19.64 | 199.00 | 0.7295 |
| | | RepQ* | 645.72 | 5.20 | 19.87 | 254.70 | 0.7929 |
| 100 | | **Ours** | 645.72 | **4.73** | **17.83** | **277.27** | **0.8270** |
| | | PTQ4DM | 323.79 | 89.78 | 57.20 | 26.02 | 0.2146 |
| | | Q-Diffusion | 323.79 | 54.95 | 36.13 | 42.80 | 0.3846 |
| | 4/8 | PTQD | 323.79 | 55.96 | 37.24 | 42.87 | 0.3948 |
| | | RepQ* | 323.79 | 26.64 | 29.42 | 91.39 | 0.4347 |
| | | **Ours** | 323.79 | **7.75** | **22.01** | **190.38** | **0.7292** |
| | 32/32 | FP | 2575.42 | 6.02 | 21.77 | 246.24 | 0.7812 |
| | | PTQ4DM | 645.72 | 17.52 | 84.28 | 154.08 | 0.6574 |
| | | Q-Diffusion | 645.72 | 14.61 | 27.57 | 153.01 | 0.6601 |
| | 8/8 | PTQD | 645.72 | 15.21 | 27.52 | 151.60 | 0.6578 |
| | | RepQ* | 645.72 | 7.17 | 23.67 | 224.83 | 0.7496 |
| 50 | | **Ours** | 645.72 | **5.45** | **19.50** | **250.68** | **0.7882** |
| | | PTQ4DM | 323.79 | 102.52 | 58.66 | 19.29 | 0.1710 |
| | | Q-Diffusion | 323.79 | 22.89 | 29.49 | 109.22 | 0.5752 |
| | 4/8 | PTQD | 323.79 | 25.62 | 29.77 | 104.28 | 0.5667 |
| | | RepQ* | 323.79 | 31.39 | 30.77 | 80.64 | 0.4091 |
| | | **Ours** | 323.79 | **9.17** | **24.29** | **179.95** | **0.7052** |

Table 2: Performance on ImageNet 512×512 with W4A8.

| Timesteps | Method | FID ↓ | sFID ↓ | IS ↑ | Precision ↑ |
|---|---|---|---|---|---|
| | FP | 8.39 | 36.25 | 257.06 | 0.8426 |
| | PTQ4DM | 68.43 | 57.76 | 35.16 | 0.4712 |
| 250 | QDiffusion | 58.81 | 56.75 | 31.29 | 0.4878 |
| | PTQD | 87.53 | 74.55 | 34.40 | 0.5144 |
| | RepQ* | 59.65 | 73.71 | 33.19 | 0.3676 |
| | **Ours** | **17.55** | **46.92** | **123.49** | **0.7592** |
| | FP | 9.06 | 37.58 | 239.03 | 0.8300 |
| | PTQ4DM | 70.63 | 57.73 | 33.82 | 0.4574 |
| 100 | QDiffusion | 62.05 | 57.02 | 29.52 | 0.4786 |
| | PTQD | 81.17 | 66.58 | 35.67 | 0.5166 |
| | RepQ* | 62.70 | 73.29 | 31.44 | 0.3606 |
| | **Ours** | **19.00** | **50.71** | **121.35** | **0.7514** |
| | FP | 11.28 | 41.70 | 213.86 | 0.8100 |
| | PTQ4DM | 71.69 | 59.10 | 33.77 | 0.4604 |
| 50 | QDiffusion | 53.49 | **50.27** | 38.99 | 0.5430 |
| | PTQD | 73.45 | 59.14 | 39.63 | 0.5508 |
| | RepQ* | 65.92 | 74.19 | 30.92 | 0.3542 |
| | **Ours** | **19.71** | 52.27 | **118.32** | **0.7336** |

We reimplement these methods to suit the unique structure of DiTs. Considering the architectural similarity between DiTs and ViTs [11], our analysis also includes RepQ-ViT [21], the state-of-the-art PTQ method initially designed for ViTs. We enhance RepQ-ViT (denoted as RepQ*) by extending the calibration set to integrate temporal dynamics and customizing its advanced channel-wise and $\log\sqrt{2}$ quantizers specifically for DiTs.

Tables 1 and 2 report the outcomes on large-scale class-conditional image generation for ImageNet 256×256 and 512×512, respectively. Table 1 demonstrates the effectiveness of PTQ4DiT across various quantization settings and timesteps. Notably, our finding reveals that at 8-bit precision (W8A8), PTQ4DiT closely matches the generative capabilities of the

FP models, whereas most baseline methods experience significant performance losses. At the more stringent 4-bit weight precision (W4A8), all baseline methods exhibit more considerable degradation. For instance, under 250 timesteps, PTQ4DM [44] sees a drastic FID increase of 68.05. In contrast, our PTQ4DiT only incurs a slight increase of 2.56. This resilience remains evident as the number of timesteps decreases, underscor-

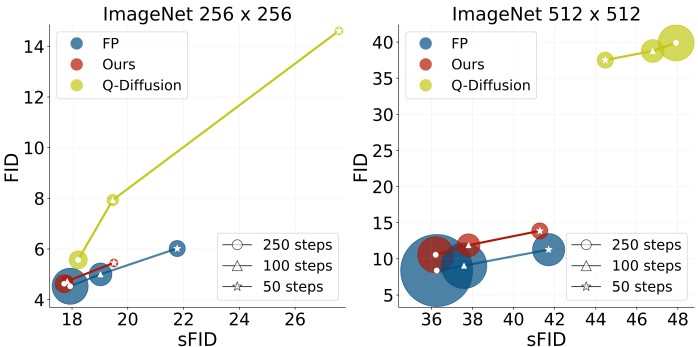

Figure 6: Quantization performance on W8A8. The circle size represents the computational load (in Gflops).

ing the robustness of PTQ4DiT in resource-limited environments. Moreover, PTQ4DiT markedly outperforms mainstream methods at the higher $512\times512$ resolution, further validating its superiority. For example, using 250 timesteps, PTQ4DiT substantially lowers FID by 41.26 and sFID by 9.83 over the second-best method, Q-Diffusion. Figure 6 depicts the efficiency-vs-efficacy trade-off on W8A8 across various timestep configurations. Our PTQ4DiT achieves comparable performance levels to FP models but with considerably reduced computational costs, offering a viable alternative for high-quality image generation. Figures 5, 8, and 9 also present randomly generated images for visual comparisons, highlighting PTQ4DiT's ability to produce images of superior quality.

## 5.3  Ablation Study

To verify the efficacy of CSB and SSC, we conduct an ablative study on the challenging W4A8 quantization. Experiments are performed on ImageNet $256\times256$ using 250 sampling timesteps. Three method variants are considered in our ablation: **(i) Baseline**, which applies basic linear quantization on DiTs, **(ii) Baseline + CSB**, which integrates CSB in the linear layers within MHSA and PF modules, where the Salience Balancing Matrices $\mathbf{B^X}$ and $\mathbf{B^W}$ are estimated based on distributions at the midpoint timestep $\frac{T}{2}$, and **(iii) Baseline + CSB + SSC**, which is the complete PTQ4DiT. Results detailed in Table 3 indicate that each proposed component improves the performance, validating their effectiveness. Particularly, CSB enhances upon the Baseline by a large margin, decreasing FID by 14.37 and sFID by 2.35, suggesting its critical role in alleviating the severe quantization difficulties inherent in DiTs. Note that with the addition of CSB, our method surpasses Q-Diffusion [18], a leading PTQ method for diffusion models. Moreover, integrating SSC further boosts our PTQ4DiT towards state-of-the-art performance, facilitating high-quality image generation at W4A8 precision, as shown in Figure 5.

Table 3: Ablation study on ImageNet $256\times256$ with W4A8.

| Method | Size (MB) | FID ↓ | sFID ↓ | IS ↑ | Precision ↑ |
|---|---|---|---|---|---|
| FP | 2575.42 | 4.53 | 17.93 | 278.50 | 0.8231 |
| Q-Diffusion | 323.79 | 15.31 | 26.04 | 134.71 | 0.6194 |
| Baseline | 323.79 | 22.54 | 27.31 | 105.55 | 0.4791 |
| + CSB | 323.79 | 8.17 | 24.96 | 187.94 | 0.7183 |
| **+ CSB + SSC (Ours)** | 323.79 | **7.09** | **23.23** | **201.91** | **0.7217** |

## 6  Conclusion

This paper proposes **PTQ4DiT**, a novel Post-training Quantization (PTQ) method for Diffusion Transformers (DiTs). Our analysis identifies the primary challenges in effective DiT quantization: the pronounced quantization errors incurred by salient channels with extreme magnitudes and the temporal variability in salient activation. To address these challenges, we design **C**hannel-wise **S**alience **B**alancing (**CSB**) and **S**pearman's $\rho$-guided **S**alience **C**alibration (**SSC**). Specifically, CSB utilizes the complementarity nature of salient channels to redistribute the extremes within activations and weights toward the balanced salience. SSC dynamically adjusts salience evaluations across different timesteps, prioritizing timesteps where salient activation and weight channels exhibit significant complementarity, thereby mitigating overall quantization difficulties. To avoid extra

computational costs of PTQ4DiT, we also devise a re-parameterization strategy for efficient inference. Experiments show that our PTQ4DiT can effectively quantize DiTs to 8-bit precision (W8A8) and further advance to 4-bit weight (W4A8) while maintaining high-quality image generation capabilities.

**Acknowledgements.** This research is supported by NSF IIS-2309073 and ECCS-2123521. This article solely reflects the opinions and conclusions of authors and not funding agencies.

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

# A  Structures of MHSA and PF with Adjusted Linear Layers

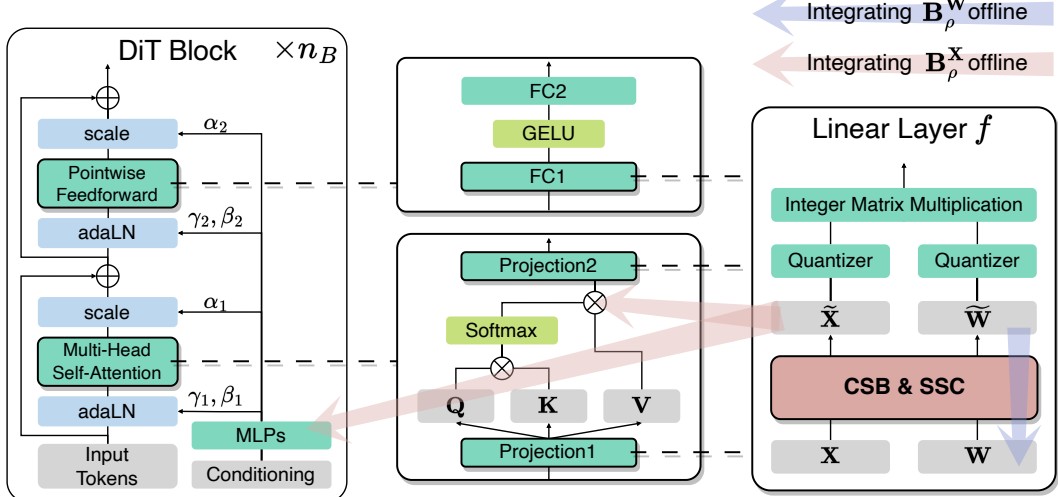

Figure 7: Illustration of structures of the MHSA and PF modules within DiT Blocks [37]. Our proposed CSB and SSC are embedded in their linear layers, including Projection1, Projection2, and FC1. CSB and SSC collectively mitigate the quantization difficulties by transforming both activations and weights using Salience Balancing Matrices, $\mathbf{B}_\rho^{\mathbf{W}}$ and $\mathbf{B}_\rho^{\mathbf{X}}$. To prevent extra computational burdens at inference time, $\mathbf{B}_\rho^{\mathbf{W}}$ is absorbed into the weight matrix of the linear layer $f$. Meanwhile, $\mathbf{B}_\rho^{\mathbf{X}}$ is integrated offline into the MLPs layer prior to adaLN modules for Projection1 and FC1, and into the preceding matrix multiplication operation for Projection2.

The Multi-Head Self-Attention (MHSA) and Pointwise Feedforward (PF) modules are essential for processing input tokens and conditional information in DiT Blocks [37]. As depicted in Figure 7, we incorporate our Channel-wise Salience Balancing (CSB) and Spearman's $\rho$-guided Salience Calibration (SSC) techniques into the linear layers within both modules. These techniques are designed to mitigate the quantization difficulties by dynamically adjusting the salience of activations and weights via Salience Balancing Matrices. Through the adjustments, CSB and SSC allow for more uniform distributions of activation and weight magnitudes across salient channels, reducing the impact of extreme values and enhancing the overall stability of the quantization process.

To eliminate additional computational demands during inference, the Salience Balancing Matrices, $\mathbf{B}_\rho^{\mathbf{W}}$ and $\mathbf{B}_\rho^{\mathbf{X}}$, are pre-integrated into the DiT Blocks. Specifically, we replace the weight matrix of the linear layer $f$ with $\widetilde{\mathbf{W}} = \mathbf{B}_\rho^{\mathbf{W}}\mathbf{W}$ and integrate $\mathbf{B}_\rho^{\mathbf{X}}$ into the preceding linear transformations. For Projection1 and FC1, $\mathbf{B}_\rho^{\mathbf{X}}$ is absorbed into the MLPs before the adaLN modules, while for Projection2, it can be absorbed within the matrix multiplication [12, 53, 43]. Derivations of the integration are provided in Appendix D.

# B  PTQ4DiT Pipeline

This section provides a comprehensive description of the PTQ4DiT pipeline, detailed in Algorithm 1. The PTQ4DiT is designed to enhance the performance of quantized DiTs by addressing quantization challenges through sophisticated salience estimation and balancing strategies.

The full algorithm mainly consists of five steps: ❶ The pipeline begins with estimating activation and weight salience for the pre-trained model using a calibration dataset. ❷ Following the estimation, we employ Spearman's $\rho$-guided Salience Calibration to compute correlation coefficients between activation salience and weight salience, which helps determine the weighting factors for each timestep. These factors are crucial for computing a temporally adjusted salience, which aims to minimize quantization errors that typically occur due to misalignment in salience peaks across the timesteps. ❸ The Channel-wise Salience Balancing step follows, wherein Salience Balancing Matrices are constructed to redistribute the activation and weight values channel-wise. Specifically, for each

---

**Algorithm 1** Post-Training Quantization for Diffusion Transformers (PTQ4DiT)

---

1: **Input:** Pre-trained DiT model, Activation sequence $\mathbf{X}^{(1:T)}$ from calibration dataset
2: **Output:** Quantized DiT model with low-bit activations and weights
3: ❶ **Preparation:**
4: Estimate activation salience $\mathbf{s}(\mathbf{X}^{(t)})$ at each timestep $t$      ▷ Using Eq. (9)
5: Estimate weight salience $\mathbf{s}(\mathbf{W})$      ▷ Using Eq. (9)
6: ❷ **Spearman's $\rho$-guided Salience Calibration:**
7: Compute correlation coefficients $\{\rho(\mathbf{s}(\mathbf{X}^{(t)}), \mathbf{s}(\mathbf{W}))\}_{t=1}^{T}$
8: Compute weighting factors $\{\eta_t\}_{t=1}^{T}$      ▷ Using Eq. (11)
9: Compute temporal salience $\mathbf{s}_\rho(\mathbf{X}^{(1:T)})$      ▷ Using Eq. (10)
10: ❸ **Channel-wise Salience Balancing:**
11: Compute balanced salience $\widetilde{s}_\rho(\mathbf{X}_j^{(1:T)}, \mathbf{W}_j)$ for each channel $j$      ▷ Using Eqs. (5), (14)
12: Construct refined Salience Balancing Matrices $\mathbf{B}_\rho^{\mathbf{X}}$ and $\mathbf{B}_\rho^{\mathbf{W}}$      ▷ Using Eqs. (6), (7)
13: ❹ **Re-Parameterization:**
14: Integrate $\mathbf{B}_\rho^{\mathbf{W}}$ into the weight matrix of linear layers offline      ▷ By $\widetilde{\mathbf{W}} = \mathbf{B}_\rho^{\mathbf{W}} \mathbf{W}$
15: Integrate $\mathbf{B}_\rho^{\mathbf{X}}$ into the MLPs before adaLN modules offline      ▷ Using Eqs. (13), (20)
16: ❺ **Quantization:**
17: Obtain $\widetilde{\mathbf{X}} = \mathbf{X}\mathbf{B}_\rho^{\mathbf{X}}$ and $\widetilde{\mathbf{W}} = \mathbf{B}_\rho^{\mathbf{W}}\mathbf{W}$ without extra computational demand during inference
18: Perform quantization on balanced activation $\widetilde{\mathbf{X}}$ and weight $\widetilde{\mathbf{W}}$

---

channel $j$, the balanced salience $\widetilde{s}_\rho(\mathbf{X}_j^{(1:T)}, \mathbf{W}_j)$ is given by:

$$\widetilde{s}_\rho(\mathbf{X}_j^{(1:T)}, \mathbf{W}_j) = (s_\rho(\mathbf{X}_j^{(1:T)}) \cdot s(\mathbf{W}_j))^{\frac{1}{2}}, \tag{14}$$

where $s_\rho(\mathbf{X}_j^{(1:T)})$ is the $j$-th element of $\mathbf{s}_\rho(\mathbf{X}^{(1:T)})$. Then, we formulate the refined Salience Balancing Matrices $\mathbf{B}_\rho^{\mathbf{X}}$ and $\mathbf{B}_\rho^{\mathbf{W}}$ based on $\widetilde{s}_\rho(\mathbf{X}_j^{(1:T)}, \mathbf{W}_j)$ and $s_\rho(\mathbf{X}_j^{(1:T)})$, as detailed in Eq. (15). This step is pivotal in aligning the activation and weight distributions, thereby minimizing the overall quantization difficulty. ❹ In the Re-Parameterization phase, these balancing matrices are integrated into the pre-trained model, ensuring that no additional computational cost is required during inference. This integration maintains computational efficiency while retaining the benefits of our salience balancing technique. ❺ Finally, we perform quantization on the model with balanced activations and weights, setting the stage for the deployment of efficient and effective quantized DiTs in resource-constrained environments.

## C  Proof of Mathematical Equivalence

In this section, we provide detailed proof demonstrating that our PTQ4DiT maintains mathematical equivalence to the original linear layers. This proof ensures that the balancing operation does not alter the original computational outcomes of the full-precision models.

In PTQ4DiT, we introduce the Salience Balancing Matrices $\mathbf{B}_\rho^{\mathbf{X}}$ and $\mathbf{B}_\rho^{\mathbf{W}}$, which are diagonal matrices intended to balance the salience across activation and weight channels. We verify the inverse relationship of $\mathbf{B}_\rho^{\mathbf{X}}$ and $\mathbf{B}_\rho^{\mathbf{W}}$ mentioned in Section 4.3:

$$
\begin{aligned}
&\mathbf{B}_\rho^{\mathbf{X}} \cdot \mathbf{B}_\rho^{\mathbf{W}} \\
&= \mathrm{diag}\left(\frac{\widetilde{s}_\rho(\mathbf{X}_1^{(1:T)}, \mathbf{W}_1)}{s_\rho(\mathbf{X}_1^{(1:T)})}, \ldots, \frac{\widetilde{s}_\rho(\mathbf{X}_{d_{in}}^{(1:T)}, \mathbf{W}_{d_{in}})}{s_\rho(\mathbf{X}_{d_{in}}^{(1:T)})}\right) \cdot \mathrm{diag}\left(\frac{\widetilde{s}_\rho(\mathbf{X}_1^{(1:T)}, \mathbf{W}_1)}{s(\mathbf{W}_1)}, \ldots, \frac{\widetilde{s}_\rho(\mathbf{X}_{d_{in}}^{(1:T)}, \mathbf{W}_{d_{in}})}{s(\mathbf{W}_{d_{in}})}\right) \\
&= \mathrm{diag}\left(\frac{\widetilde{s}_\rho(\mathbf{X}_1^{(1:T)}, \mathbf{W}_1)}{s_\rho(\mathbf{X}_1^{(1:T)})} \cdot \frac{\widetilde{s}_\rho(\mathbf{X}_1^{(1:T)}, \mathbf{W}_1)}{s(\mathbf{W}_1)}, \ldots, \frac{\widetilde{s}_\rho(\mathbf{X}_{d_{in}}^{(1:T)}, \mathbf{W}_{d_{in}})}{s_\rho(\mathbf{X}_{d_{in}}^{(1:T)})} \cdot \frac{\widetilde{s}_\rho(\mathbf{X}_{d_{in}}^{(1:T)}, \mathbf{W}_{d_{in}})}{s(\mathbf{W}_{d_{in}})}\right) \\
&= \mathrm{diag}\left(\frac{(s_\rho(\mathbf{X}_1^{(1:T)}) \cdot s(\mathbf{W}_1))^{\frac{1}{2} \cdot 2}}{s_\rho(\mathbf{X}_1^{(1:T)}) \cdot s(\mathbf{W}_1)}, \ldots, \frac{(s_\rho(\mathbf{X}_{d_{in}}^{(1:T)}) \cdot s(\mathbf{W}_{d_{in}}))^{\frac{1}{2} \cdot 2}}{s_\rho(\mathbf{X}_{d_{in}}^{(1:T)}) \cdot s(\mathbf{W}_{d_{in}})}\right) = \mathbf{I},
\end{aligned}
\tag{15}
$$

where $\mathbf{I}$ denotes the identity matrix. Therefore, we can derive the mathematical equivalence:

$$\widetilde{\mathbf{X}} \cdot \widetilde{\mathbf{W}} = (\mathbf{X}\mathbf{B}_\rho^{\mathbf{X}}) \cdot (\mathbf{B}_\rho^{\mathbf{W}}\mathbf{W}) = \mathbf{X} \cdot (\mathbf{B}_\rho^{\mathbf{X}}\mathbf{B}_\rho^{\mathbf{W}}) \cdot \mathbf{W} = \mathbf{X} \cdot \mathbf{W}. \tag{16}$$

## D    Derivations of Post-adaLN Integration

This section details the integration of the Salience Balancing Matrix $\mathbf{B}_\rho^{\mathbf{X}}$ into the MLPs before the adaptive Layer Norm (adaLN) modules [37], aimed at eliminating extra computational overhead at the inference stage. Recall that the initial formulation of adaLN on the input latent noise $\mathbf{Z} \in \mathbb{R}^{n \times d_{in}}$ is given by:

$$\mathbf{X} = \mathrm{adaLN}(\mathbf{Z}) = \mathrm{LN}(\mathbf{Z}) \odot (\mathbf{1} + \boldsymbol{\gamma}) + \boldsymbol{\beta}, \tag{17}$$

where $\boldsymbol{\gamma}, \boldsymbol{\beta} \in \mathbb{R}^{d_{in}}$ are scale and shift parameters, respectively, regressed by MLPs based on the conditional input $\mathbf{c} \in \mathbb{R}^{d_{in}}$:

$$(\boldsymbol{\gamma}, \boldsymbol{\beta}) = \mathrm{MLPs}(\mathbf{c}) = \mathbf{c} \cdot (\mathbf{W}_{\boldsymbol{\gamma}}, \mathbf{W}_{\boldsymbol{\beta}}) + (\mathbf{b}_{\boldsymbol{\gamma}}, \mathbf{b}_{\boldsymbol{\beta}}). \tag{18}$$

Here, $\mathbf{W}_{\boldsymbol{\gamma}}, \mathbf{W}_{\boldsymbol{\beta}}$ are weight matrices, and $\mathbf{b}_{\boldsymbol{\gamma}}, \mathbf{b}_{\boldsymbol{\beta}}$ are bias terms. In PTQ4DiT, $\widetilde{\mathbf{X}}$ is obtained by applying $\mathbf{B}_\rho^{\mathbf{X}}$ to the output of adaLN as follows:

$$\widetilde{\mathbf{X}} = \mathbf{X}\mathbf{B}_\rho^{\mathbf{X}} = \mathrm{LN}(\mathbf{Z}) \odot (\mathbf{B}_\rho^{\mathbf{X}} + \boldsymbol{\gamma}\mathbf{B}_\rho^{\mathbf{X}}) + \boldsymbol{\beta}\mathbf{B}_\rho^{\mathbf{X}}, \tag{19}$$

which echos with Eq. (13). To avoid additional matrix multiplications in $\boldsymbol{\gamma}\mathbf{B}_\rho^{\mathbf{X}}$ and $\boldsymbol{\beta}\mathbf{B}_\rho^{\mathbf{X}}$, we can pre-absorb $\mathbf{B}_\rho^{\mathbf{X}}$ in MLPs' weights and biases offline, expressed as:

$$(\widetilde{\mathbf{W}}_{\boldsymbol{\gamma}}, \widetilde{\mathbf{W}}_{\boldsymbol{\beta}}) = (\mathbf{W}_{\boldsymbol{\gamma}}\mathbf{B}_\rho^{\mathbf{X}}, \mathbf{W}_{\boldsymbol{\beta}}\mathbf{B}_\rho^{\mathbf{X}}), \quad (\widetilde{\mathbf{b}}_{\boldsymbol{\gamma}}, \widetilde{\mathbf{b}}_{\boldsymbol{\beta}}) = (\mathbf{b}_{\boldsymbol{\gamma}}\mathbf{B}_\rho^{\mathbf{X}}, \mathbf{b}_{\boldsymbol{\beta}}\mathbf{B}_\rho^{\mathbf{X}}). \tag{20}$$

Thus, the re-parameterized MLPs can directly produce the adjusted scale and shift parameters:

$$\begin{aligned}
\widetilde{\mathrm{MLPs}}(\mathbf{c}) \\
&= \mathbf{c} \cdot (\widetilde{\mathbf{W}}_{\boldsymbol{\gamma}}, \widetilde{\mathbf{W}}_{\boldsymbol{\beta}}) + (\widetilde{\mathbf{b}}_{\boldsymbol{\gamma}}, \widetilde{\mathbf{b}}_{\boldsymbol{\beta}}) \\
&= \mathbf{c} \cdot (\mathbf{W}_{\boldsymbol{\gamma}}\mathbf{B}_\rho^{\mathbf{X}}, \mathbf{W}_{\boldsymbol{\beta}}\mathbf{B}_\rho^{\mathbf{X}}) + (\mathbf{b}_{\boldsymbol{\gamma}}\mathbf{B}_\rho^{\mathbf{X}}, \mathbf{b}_{\boldsymbol{\beta}}\mathbf{B}_\rho^{\mathbf{X}}) \\
&= (\mathbf{c} \cdot (\mathbf{W}_{\boldsymbol{\gamma}}, \mathbf{W}_{\boldsymbol{\beta}}) + (\mathbf{b}_{\boldsymbol{\gamma}}, \mathbf{b}_{\boldsymbol{\beta}})) \cdot \mathbf{B}_\rho^{\mathbf{X}} \\
&= (\boldsymbol{\gamma}, \boldsymbol{\beta}) \cdot \mathbf{B}_\rho^{\mathbf{X}} \\
&= (\boldsymbol{\gamma}\mathbf{B}_\rho^{\mathbf{X}}, \boldsymbol{\beta}\mathbf{B}_\rho^{\mathbf{X}}).
\end{aligned} \tag{21}$$

This allows for obtaining $\widetilde{\mathbf{X}}$ without extra computational burden at the inference stage.

## E    Additional Visualization Results

Figures 8 and 9 supplement visualization results of our PTQ4DiT on W8A8 quantization, compared with baseline PTQ methods and the full-precision (FP) counterpart, on ImageNet 512×512 and 256×256. Our method generates results that closely mirror those of the FP models, presenting finer details and richer semantic content than the baseline approaches.

## F    Limitations and Broader Impacts

This work introduces a pioneering solution facilitating the broad deployment of Diffusion Transformers (DiTs) through Post-training Quantization. Our method substantially lowers computational and memory demands, thereby improving the accessibility of DiTs. Currently, our research concentrates on visual generation. For future work, we plan to extend our methodology to other generative models across various modalities, such as audio and 3D. However, there remains an inherent risk that these generative models could be utilized to produce disinformation. While our study contributes to the widespread application of DiTs, it does not address such ethical risks. We recognize the importance of developing safeguards and encourage further research into strategies that can prevent the misuse of these powerful generative technologies.

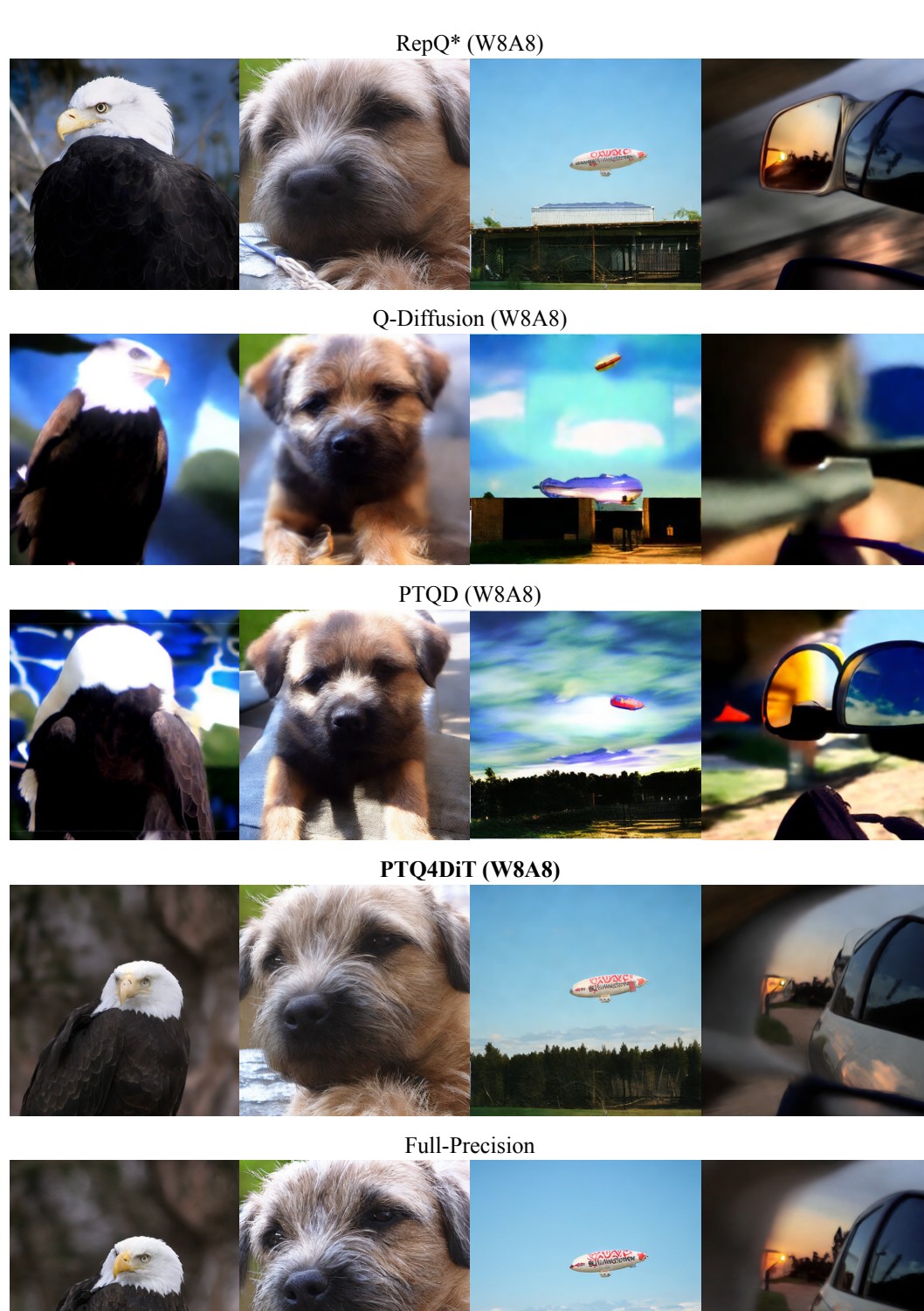

Figure 8: Random samples generated by different PTQ methods with W8A8 quantization, alongside the full-precision DiTs [37], on ImageNet 512×512.

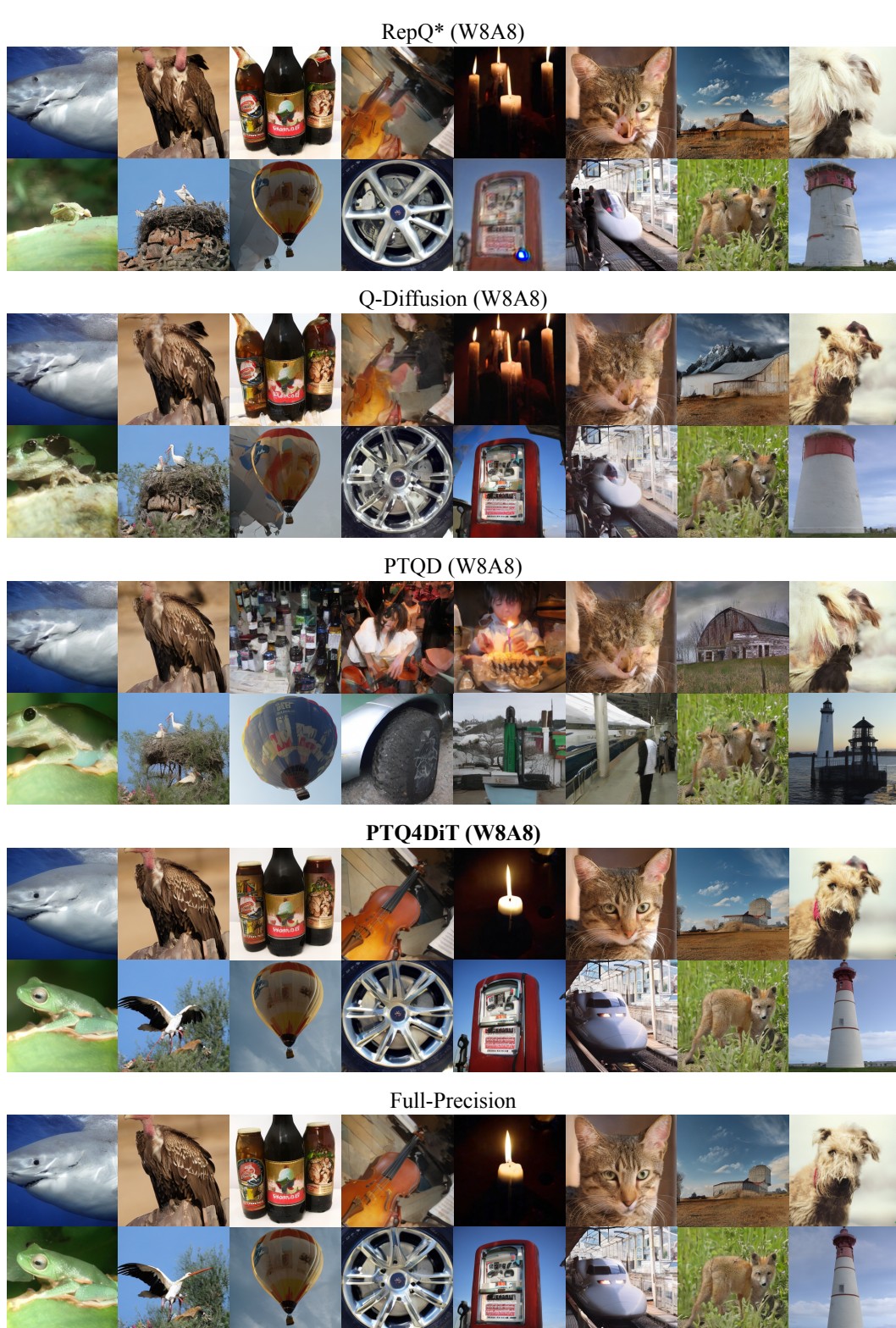

Figure 9: Random samples generated by different PTQ methods with W8A8 quantization, alongside the full-precision DiTs [37], on ImageNet 256×256.

