# OpenReview forum: "PTQ4DiT: Post-training Quantization for Diffusion Transformers"
_NeurIPS.cc/2024/Conference — NeurIPS 2024 poster_

### Official Review · Reviewer_a8C3 · 2024-06-14

**Soundness:** 3
**Presentation:** 3
**Contribution:** 3
**Rating:** 5
**Confidence:** 4

**Summary:**

This paper presents PTQ4DiT, a quantization method designed for diffusion transformers. The method focuses on addressing quantization challenges due to extreme magnitudes in salient channels and the temporal variability of activations across multiple timesteps. To combat these issues, it incorporates techniques like Channel-wise Salience Balancing (CSB) and Spearmen’s ρ-guided Salience Calibration (SSC). These strategies help in redistributing magnitudes to minimize quantization errors and adapt dynamically across different timesteps, significantly enhancing the performance of quantized DiTs without re-training the original models.

**Strengths:**

- The method incorporates temporal information into the calibration process for salience balancing.
- The experimental results are robust, covering a wide range of scenarios and effectively demonstrating the method's efficacy across different settings.

**Weaknesses:**

- The classifier-free guidance scales used for sampling are not specified, which could impact the reproducibility and evaluation of the model's performance.
- Under the W4A8 setting, the method exhibits significant degradation in performance, suggesting limitations in its effectiveness at lower bit-widths.

**Questions:**

see weaknesses

**Limitations:**

This work reasonably discussed the limitations and future work

---

> ### Author Rebuttal · Authors · 2024-08-06
>
> ## **Response to Weakness 1: Classifier-free guidance scales**
> Thank you for pointing out this important concern. We set classifier-free guidance scales as 1.5 in all the experiments in our paper, following the origin DiT work [1].
>
> [1] Scalable diffusion models with transformers. In CVPR, 2023.
>
>
> ## **Response to Weakness 2: W4A8 quantization**
> Thanks for the valuable feedback. Here, we would like to emphasize the challenging W4A8 quantization and the contribution of our PTQ4DiT.
>
> **a. Challenge of W4A8 quantization**
>
> Prior studies [1, 2] have revealed that W4A8 quantization presents non-trivial challenges due to the severe information loss in low-bit model weights and have attempted to mitigate the degradation of W4A8 U-Net-based diffusion models. However, we observed that these methods result in substantial performance loss when applied to W4A8 Diffusion Transformers (DiTs), indicating the significant difficulty of low-bit quantization for DiT architectures.
>
> **b. Contribution of PTQ4DiT**
>
> This is the first time a PTQ method has enabled high-quality generation at W4A8 bit-widths for DiTs, paving a promising path for future research in this field. Specifically, our work delves into DiT quantization and identifies two key challenges (Section 3). Then, we develop PTQ4DiT to address these challenges. Encouragingly, we find that PTQ4DiT consistently shows significant performance improvements over mainstream methods, as detailed in Tables 1 and 2 of our paper. Figure 5 further demonstrates that PTQ4DiT facilitates high-quality image generation despite the difficulty inherent in the lower bit-width of W4A8.
>
> [1] Q-Diffusion: Quantizing Diffusion Models. In ICCV, 2023.
>
> [2] PTQD: Accurate Post-Training Quantization for Diffusion Models. In NIPS, 2023.

---

> > ### Comment · Reviewer_a8C3 · 2024-08-10
> >
> > After reading the rebuttal and the comments from other reviewers, I keep my initial rating.

---

> ### Author Response · Authors · 2024-08-12
> **Thank you for your response**
>
> Dear Reviewer a8C3,
>
> We sincerely appreciate your prompt response and are grateful that you found our rebuttal beneficial. Thank you once more for your valuable feedback in enhancing our submission.

---

### Official Review · Reviewer_jrKh · 2024-07-11

**Soundness:** 3
**Presentation:** 3
**Contribution:** 3
**Rating:** 5
**Confidence:** 5

**Summary:**

- The paper introduces PTQ4DiT, a post-training quantization method for Diffusion Transformers. This approach can facilitate the widespread deployment of DiTs. By investigating the distribution of activation and weight of DiTs, PTQ4DiT designs the Channel-wise Balancing and timestep-aware Salience Calibration. Through these, PTQ4DiT effectively quantizes DiTs to W4A8 and reduce computational costs while maintaining image generation quality.

**Strengths:**

- The paper identifies the two key challenges associated with quantizing DiTs and provides simple but effective methodology for effective DiT quantization.
- The description and illustration of the paper is clear and easy to follow. Detailed description of the reparamterization process of the quantization parameters are provide

**Weaknesses:**

- **Effectiveness of Salience Calibration:** The methodology sections 4.1 and 4.3 of the paper  resembles existing work on quantization [1]. The primary novel contribution of this paper is the 'Salience Calibration' technique, which addresses the specific challenge of activation timestep-variance in DiTs. It is critical to verify its importance. However, the evidence supporting the effectiveness of the Salience Calibration (SSC) is currently confined to Table 3, which presents results for ImageNet with 256x256 resolution, using a W4A8 configuration. The improvement observed with SSC is only moderate when compared to the CSB method. To more effectively emphasize the paper's unique contribution, additional evidence demonstrating the effectiveness of CSB would be beneficial."
- **Overheads for Salience Calibration:** The 'Salience Calibration' technique introduces a timestep-wise correction coefficient to refine the activation $S(X(t))$ and weight $S(W)$ distributions, resulting in the adjusted binary matrices $B_{\rho}^x$ and $B_{\rho}^w$. This refinement implies that $B_{\rho}^w$ may differ for each timestep t. However, the weights are reparameterized offline prior to quantization. Introducing a time-varying $B_{\rho}^w$ could lead to the generation of distinct quantized integer (INT) weights for each timestep, which may complicate efforts to reduce the model's weight memory cost. This could involve either storing multiple sets of INT weights or repeatedly offloading weights for each timestep.

[1] Xiao, Guangxuan, et al. "Smoothquant: Accurate and efficient post-training quantization for large language models." International Conference on Machine Learning. PMLR, 2023

**Questions:**

- Is the 'Spearman’s ρ-guided Salience Calibration (SSC)' method capable of generalizing across a different timesteps and solvers? It seems that the calibration process depends on the number of timesteps. Does this imply that recalibration is necessary when employing SSC with different models, timesteps, or solvers?
- In the context of reparameterization, the paper specifically addresses the 'Post-adaLN' and 'Post-Matrix-Multiplication' scenarios. However, within the Feed-Forward Network (FFN) layers of DiTs, there are instances where two consecutive linear layers do not conform to either of these cases. It raises the question of how to approach reparameterization in such circumstances."
- Some unclear details about the quantization process:
  - Are all layers in the DiTs quantized? e.g., time embedding linear layers, attention matmuls.
  - In Line 237, the authors state that " the optimization of quantization parameters follows the implementation of Q-Diffusion". Does this mean that the quantization process involves gradient-based optimization of scaling factors, and Adaround of zero-points? What is the cost of the PTQ process.
- The statement “the lower the correlation between activation salience s(X(t)) and weight salience s(W), the greater the reduction effect in overall channel salience”  in Sec. 4.2 should be more clearly explained. What is the definition of "correlation salience"?

**Limitations:**

- The authors have discussed the limitations in the appendix.

---

> ### Author Rebuttal · Authors · 2024-08-06
>
> ## **Response to Weakness 1: Effectiveness of SSC**
> We clarify the innovation of CSB and provide additional evidence of the efficacy of SSC.
>
> **a. Innovation of CSB**
>
> While CSB shares the general concept of distribution re-scaling and re-parameterization with Smoothquant, it has unique innovation. Our CSB is designed for quantizing DiTs, whereas existing methods, such as Smoothquant, focus on quantizing LLMs. The quantization of DiTs presents unique challenges due to the extreme values in **both activations and weights**. In contrast, Smoothquant identifies outliers **only in activations** within LLMs and addresses this by migrating outliers from activations to weights. Fortunately, the extreme values do not occur in the same channels of activation and weight in DiTs. Our CSB is then derived from this complementarity to balance the salience between activation and weight, alleviating quantization errors for both.
>
> **b. More SSC ablation**
>
> To validate the importance of SSC, we conduct additional experiment on the challenging ImageNet 512x512 with W4A8 using 50 timesteps:
> | Method | FID ↓ | sFID ↓ | IS ↑ | Precision↑|
> |---|---|---|---|---|
> |Baseline|56.94|74.55|39.16|0.4178|
> |+CSB|25.45|58.05|90.73|0.6390|
> |+CSB+SSC|19.71|52.27|118.32|0.7336|
>
> The results demonstrate the significant impact of SSC, as it further enhances the benefits introduced by CSB.
>
> **c. Direct evidence of SSC's effect**
>
> Please see **General Response 2: Experiment on direct evidence**.
>
>
> ## **Response to Weakness 2: Overheads**
> We clarify that $B_\rho^X$ and $B_\rho^W$ are not time-varying. As defined in Eq. (10), $s_\rho(X^{(1:T)})$ aggregates activation saliences across all timesteps. Therefore, the resulting refined salience is a function of the number of timesteps $T$ and does not depend on any specific timestep $t$. We then formulate $B_\rho^X$ and $B_\rho^W$ based on $s_\rho(X^{(1:T)})$ and $s(W)$, which are not time-varying.
>
> Thus, we only perform reparameterization **once before quantization, without additional memory cost for model weights**.
>
>
> ## **Response to Question 1: Recalibration**
> While SSC is orthogonal to solvers and models, it does depend on the number of timesteps ($T$ in Eq. (10)) due to its design of aggregating channel salience across T. Encouragingly, SSC is capable of generalizing across different T without recalibration. To verify its generalizability, we conduct two sets of experiments on ImageNet 256x256 with W8A8:
>
> **(I) Downsampling**: We calibrate and quantize the model using PTQ4DiT with 250 timesteps and evaluate this model using 100 and 50 timesteps.
>
> **(II) Upsampling**: We calibrate and quantize the model using PTQ4DiT with 50 timesteps and evaluate this model using 100 and 250 timesteps.
>
> |Calibration Timesteps|Evaluation Timesteps|FID↓|sFID↓|IS↑|Precision↑|
> |---|---|---|---|---|---|
> |250|250|4.63 |17.72|274.86|0.8299|
> |100|100|4.73|17.83|277.27|0.8270|
> |50|50|5.45|19.50|250.68|0.7882|
> |**250**|**100**|4.86|17.76|269.93|0.8221|
> |**250**|**50**|5.47|19.32|249.01|0.7899|
> |**50**|**100**|4.99|18.12|261.84|0.8239|
> |**50**|**250**|4.73|17.89| 266.04|0.8312|
>
> The results show that both downsampling and upsampling does not significantly effect the performance, demonstrating the generalizability of SSC. We conjecture that this is due to two factors:
>
> **(I) Inherent generalizability of DiTs.** Previous studies [1,2,3] indicated that diffusion models possess strong ability to generalize across timesteps. Consequently, models trained with 250 timesteps can be directly applied to 100 and 50 timesteps with acceptable performance degradation. This interpolation ability is also evidenced by the performance of FP DiTs with 100 and 50 timesteps in Tables 1 and 2 of our paper (note that the original DiT work [4] only released the FP model for 250 timesteps).
>
> **(II) Effect of PTQ.** PTQ is formulated as a numerical problem and does not re-train the original models, which will not significantly affect the inherent interpolation ability if the quantization error remains low enough.
>
> [1] Diffusion Models Beat GANs on Image Synthesis. NIPS 2021.
>
> [2] Improved denoising diffusion probabilistic models. ICML 2021.
>
> [3] Post-training quantization on diffusion models. CVPR 2023.
>
> [4] Scalable diffusion models with transformers. CVPR 2023.
>
> ## **Response to Question 2: Reparameterization**
> We address 3 types of linear layers exhibiting significant channel salience: FC1, Projection1, and Projection2. We design reparameterization strategies for these layers. Specifically, FC1 follows the 'Post-adaLN' scenario. Thus, the Balancing Matrices can be integrated to the adaLN before Pointwise Feedforward. We further absorb them into MLPs regressing $\gamma_2$ and $\beta_2$, a process detailed in Appendix D.
>
> We illustrate the integration strategies in Figure 7 and discuss them in the caption. We apologize for causing any confusion.
>
> ## **Response to Question 3: Quantization details**
> **a. Are all layers in the DiTs quantized?**
>
> We quantize all layers in the DiT models. The time embedding linear layers and attention matmuls are quantized.
>
> **b. Quantization optimization**
>
> Following Q-Diffusion, we involve gradient-based optimization for the quantization parameters, which has similar PTQ cost as that of Q-Diffusion (about 19 hours on a single NVIDIA A6000 GPU) since we do not introduce additional parameters.
>
> ## **Response to Question 4: Explaining the statement**
> Correlation salience refers to the degree of alignment between activation salience $s(X^{(t)})$ and weight salience $s(W)$. A low correlation indicates that channels with extreme values in activations are not the same as those in weights, and vice versa.
>
> Leveraging this complementarity, we propose SSC to prioritize timesteps with lower correlation between activation and weight saliences. This approach helps distribute the quantization impact more evenly across channels and timesteps, improving overall quantization performance.

---

> > ### Comment · Reviewer_jrKh · 2024-08-08
> >
> > Thank the authors for the clarification, most of my concerns are addressed. I keep my scoring as acceptance.

---

> ### Author Response · Authors · 2024-08-12
> **Thanks for your response**
>
> Dear Reviewer jrKh,
>
> Thank you for your thorough review and constructive suggestion. We are grateful for your acknowledgment of our efforts to address the concerns. Your expertise has significantly contributed to the enhancement of our work.

---

### Official Review · Reviewer_USPH · 2024-07-11

**Soundness:** 3
**Presentation:** 3
**Contribution:** 3
**Rating:** 6
**Confidence:** 4

**Summary:**

This paper proposes the first post-training quantization (PTQ) method for Diffusion Transformer (DiT). It addresses the presence of salient channels with extreme magnitudes and the temporal variability in the distributions of salient activations over multiple timesteps. Experimental results demonstrate comparable performance in low-bit scenarios.

**Strengths:**

1. This paper is well-organized and includes clear illustrations.
2. It presents the first post-training quantization method for Diffusion Transformers.
3. In the W8A8 scenario, PTQ4DiT achieves lossless performance compared to full precision.
4. The paper also includes theoretical analysis.

**Weaknesses:**

1. In line 158, the author proposes an important property: large values do not coincide in the same channels of activation and weight. This property is crucial for the proposed method but is only briefly illustrated by a sketch. We believe that such an important property should be demonstrated with statistical experiments.
2. In line 170, the author attempts to use the geometric mean to balance activation and weight channels. Is this approach well-designed? Could another type of mean be used?
3. PTQ4DiT is designed for Diffusion Transformers and is tested on traditional DiT. However, many new DiT-based models, such as pixart-alpha [1], pixart-sigma [2], SD3 [3], and Lumina [4], have emerged. To verify the generality of the method, it should be tested on a broader range of DiT-based models.
4. This method appears to work only with linear layers.

[1] PixArt-$\alpha$: Fast Training of Diffusion Transformer for Photorealistic Text-to-Image Synthesis, ICLR24.
[2] PixArt-\Sigma: Weak-to-Strong Training of Diffusion Transformer for 4K Text-to-Image Generation, arXiv.
[3] Scaling Rectified Flow Transformers for High-Resolution Image Synthesis, arXiv.
[4] Lumina-T2X: Transforming Text into Any Modality, Resolution, and Duration via Flow-based Large Diffusion Transformers, arXiv.

**Questions:**

Please refer to the weaknesses part.

**Limitations:**

Yes. The authors have addressed the limitations and broader impacts.

---

> ### Author Rebuttal · Authors · 2024-08-06
>
> Thank you very much for your acknowledgement of our work and your constructive feedbacks and suggestions.
> ## **Response to Weakness 1: Statistical experiments for the important property**
> We validate the important complementarity property by additional statistical experiments on the well-established Jaccard similarity [1].
>
> **a. Jaccard similarity**
>
> To quantitatively demonstrate the **complementarity** property (large values do not coincide in the same channels of activation and weight), we measure the **Jaccard similarity** [1] of salient activation and weight channels:
> $$
> Jaccard(S_A,S_W)=\frac{|S_A\cap S_W|}{|S_A\cup S_W|}\in[0,1],
> $$
> where $S_A$ is the set of indices of salient activation channels, $S_W$ is the set of indices of salient weight channels, and $|S|$ calculates the number of elements in set S. In this way, a lower $Jaccard(S_A, S_W)$ reflects stronger complementarity.
>
> **b. Detecting salient channels**
>
> To accurately construct the sets $S_A$ and $S_W$, we utilize a robust statistical outlier detector, **Interquartile Range (IQR)** [2], to identify the salient channels among all channels. This method identifies data points that lie significantly outside the middle 50% of the distribution as outliers. We perform IQR detection on channels' maximal absolute values to identify the salient channels.
>
> **c. Statistical experiment**
>
> We randomly select 100 samples for ImageNet 256x256 generation with total timestep T = 250. We evaluate the average Jaccard similarities at t = $\frac{1}{4}T, \frac{1}{2}T,$ and $\frac{3}{4}T$. The results are averaged over 100 samples and linear layers in DiTs:
> |Timestep t|$\frac{1}{4}T$|$\frac{1}{2}T$|$\frac{3}{4}T$|
> |---|---|---|---|
> |**$$Jaccard(S_A, S_W)$$**|0.0675|0.1364|0.1006|
>
> Our finding are:
>
> **(I)** We obtain relatively low Jaccard similarities (note that $Jaccard \in [0, 1]$), suggesting a significant complementarity between salient activation and weight channels. The complementarity property motivates our Channel-wise Salience Balancing (CSB) to redistribute extreme values among non-overlapping salient channels.
>
> **(II)** Different timesteps t exhibit various Jaccard similarities, aligning with the key observation in our paper: the temporal variation in salient channels. Such variability further indicates the necessity of our proposed Salience Calibration method (SSC) for various timesteps.
>
> [1] Distance between sets. Nature 1971.
>
> [2] Outlier detection: Methods, models, and classification. ACM Computing Surveys (CSUR) 2020.
>
>
> ## **Response to Weakness 2: Is the geometric mean well-designed?**
> **a. The effectiveness of geometric mean**
>
> The geometric mean was selected due to its suitability for the multiplicative relationship between activations and weights (Eq. (3) in our paper) and its ability to balance and minimize the influence of extremely large values, which is critical in our quantization method.
>
> **b. The uniqueness of geometric mean**
>
> Mathematically, alternative means could include the arithmetic, quadratic, or harmonic mean. However, interestingly, we found that the geometric mean is uniquely capable of ensuring the mathematical equivalence of the balancing transformation (as expressed by Eqs. (12), (15), (16) in our paper). Here, we provide a brief demonstration:
>
> $$
> \quad\frac{mean(s(X_j), s(W_j))}{s(X_j)}\cdot \frac{mean(s(X_j), s(W_j))}{s(W_j)} = 1
> $$
> $$
> \Leftrightarrow
> mean(s(X_j), s(W_j)) = (s(X_j) \cdot s(W_j))^{\frac{1}{2}}
> $$
>
> This essential property offers the feasibility of reparameterization, thereby eliminating extra computation overhead of the balancing transformation during inference.
>
> ## **Response to Weakness 3: Generality of PTQ4DiT**
> Please refer to **General Response 1: Generality of PTQ4DiT**.
>
> ## **Response to Weakness 4: Work only with linear layers**
> **a. Extending PTQ4DiT to other structures**
>
> Although our method is derived from the quantization of linear layers, the underlying idea of balancing transformation can be seamlessly extended to other structures, such as convolutional layers. This can be achieved by reformulating the convolution operation into a matrix multiplication between activations and weights [1], which is common in the practical implementation of convolution [2].
>
> **b. The reason for focusing on linear layers**
>
> The linear layers in Transformers incur significant computational and memory overhead due to the large matrix multiplications [3, 4, 5, 6], representing the primary efficiency bottleneck in Transformer models including Diffusion Transformers (DiTs).
>
> [1] Solving Oscillation Problem in Post-Training Quantization Through a Theoretical Perspective. CVPR 2023.
>
> [2] Optimizing hardware accelerated general matrix-matrix multiplication for cnns on fpgas. Transactions on Circuits and Systems 2020.
>
> [3] PTQ4ViT: Post-training quantization for vision transformers with twin uniform quantization. ECCV 2022.
>
> [4] RepQ-ViT: Scale Reparameterization for Post-Training Quantization of Vision Transformers. ICCV 2023.
>
> [5] OmniQuant: Omnidirectionally Calibrated Quantization for Large Language Models. ICLR 2024.
>
> [6] QLLM: Accurate and Efficient Low-Bitwidth Quantization for Large Language Models. ICLR 2024.

---

> > ### Comment · Reviewer_USPH · 2024-08-13
> >
> > Thank you for the clarification, which addressed most of my concerns. I keep my initial rating.

---

> > > ### Author Response · Authors · 2024-08-13
> > > **Thank you for your response**
> > >
> > > Dear Reviewer USPH,
> > >
> > > We deeply appreciate your response and are grateful that you found our rebuttal beneficial. Thank you once more for your constructive comments.

---

### Official Review · Reviewer_N2VF · 2024-07-13

**Soundness:** 3
**Presentation:** 3
**Contribution:** 2
**Rating:** 5
**Confidence:** 4

**Summary:**

This paper proposes a new PTQ method for DiTs. It develops a Channel-wise Salient Balancing method to suppress the outliers of linear layers in transformer blocks when applying activation quantization. Besides, it designs the Spearmen’s ρ-guided Salience Calibration to tackle the timestep dimension’s variety. It can improve the performance of the quantized models, comparing to directly applying the conventional quantization methods for Unet-based diffusion models, ViTs or traditional convolution networks to DiTs.

**Strengths:**

+This paper proposes a new method for DiT quantization, which achieves relatively well performance on W8A8 and W4A8.

**Weaknesses:**

-Lacks of experiments. This paper does not report the W8A8 results of ImageNet 512x512. Besides, it only conducts the experiment on one DiT model. The proposed method should be validated on more architectures.
-Lack of novelty. The idea of Re-Parameterization activation range has already been used in PTQ of ViT and LLM, such as Rep-Q [1] or outlier suppression+ [2].
-Lack of direct evidence. It lacks direct evidence to display the effectiveness of the proposed method, such as the visualization of activation before and after CSB and SSC to verify it actually suppress the salience channel.
-Lack of motivation. This paper proposes several modules, but there is no direct evidence to prove that these modules can solve the problems. For example, in RepQ’s, it chooses the mean value of activation range to scale the distribution, but this paper uses the equilibrium between weight and activation without explanation. Besides, it also not explains why using inverse Spear-man’s ρ statistic as the weigh.
[1]Li Z, Xiao J, Yang L, et al. Repq-vit: Scale reparameterization for post-training quantization of vision transformers. In ICCV, 2023.
[2]Wei X, Zhang Y, Li Y, et al. Outlier Suppression+: Accurate quantization of large language models by equivalent and effective shifting and scaling. In EMNLP, 2023.

**Questions:**

1. Please see the weaknesses.
2. Did the authors quantize the post-softmax layer?
4. The SOTA DiT models can achieve a FID around 3 on ImageNet, why are the results reported in this paper not aligned with the origin paper? Why did the authors choose different settings?

---

> ### Author Rebuttal · Authors · 2024-08-06
>
> ## **Weakness 1: More experiments**
> **a. W8A8 ImageNet 512x512**
>
> We evaluate PTQ4DiT on ImageNet 512x512 with W8A8, compared against strong Diffusion PTQ methods, including Q-Diffusion and PTQD:
> |Timesteps|Method|FID↓|sFID↓|IS↑|Precision↑|
> |---|---|---|---|---|---|
> |250|FP|8.39|36.25|257.06|0.8426|
> ||Q-Diffusion|39.91|47.91|60.78|0.644|
> ||PTQD|80.04|70.11|45.67|0.5636|
> ||**Ours**|**11.45**|**38.34**|**196.21**|**0.8266**|
> |100|FP|9.06|37.58|239.03|0.83|
> ||Q-Diffusion|38.77|46.77|63.82|0.6504|
> ||PTQD|90.77|79.16| 42.47|0.5392|
> ||**Ours**|**13.10**|**39.92**|**173.37**|**0.7866**|
> |50|FP|11.28|41.70|213.86|0.81|
> ||Q-Diffusion|37.51|**44.46**|69.55|0.642|
> ||PTQD|85.28|74.75|46.42|0.5458|
> ||**Ours**|**16.56**|45.52|**170.20**|**0.7944**|
>
> PTQ4DiT consistently outperforms mainstream methods across various timesteps. Figure 8 provides the images generated by PTQ4DiT in this setting, which closely mirror the FP model.
>
> **b. New DiT model**
>
> Please refer to **General Response 1: Generality of PTQ4DiT**.
> ## **Weakness 2: Novelty discussion**
> **a. The general concept of re-parameterization**
>
> The fundamental idea of re-parameterization in quantization is to absorb the extra affine transformation into adjacent layers, thereby enabling the construction of quantization-friendly distributions without incurring extra inference costs.
>
> **b. Existing works that use re-parameterization**
>
> Several quantization methods benefit from re-parameterization. For example, RepQ-ViT [1] uses scale re-parameterization to avoid the extra costs of channel-wise and $log\sqrt2$ quantizers for post-LayerNorm and post-Softmax activations in ViTs. OS+ [2] shifts and scales activations to address the asymmetry and concentration issues in LLMs and re-parameterizes them to avoid extra costs.
>
> **c. Unique innovation and contribution of PTQ4DiT**
>
> While PTQ4DiT shares the general concept of scale re-parameterization, its unique innovation lie in 3 aspects:
>
> **(I)** We explore the quantization of **DiTs**, while existing works focus on **ViTs** or **LLMs**.
>
> **(II)** We delve into the complementarity of **activations and weights**, while existing works [1,2] focus solely on **activations**. Specifically, we find that DiTs exhibit extreme values in both activations and weights, yet these extreme values do not simultaneously occur in the same activation and weight channels. We leverage this property to redistribute salience between activation and weight, alleviating quantization errors for both.
>
> **(III)** Unlike the **static** re-parameterization in existing works [1,2], PTQ4DiT **dynamically** adapts to the temporal variability of channel salience, a special characteristic of DiTs.
>
> [1] RepQ-ViT: Scale Reparameterization for Post-Training Quantization of Vision Transformers. ICCV 2023.
>
> [2] Outlier Suppression+: Accurate quantization of large language models by equivalent and optimal shifting and scaling. EMNLP 2023.
>
> ## **Weakness 3&4: Direct evidence and Motivation**
> **a. Direct evidence**
>
> Please refer to **General Response 2: Experiment on direct evidence**.
>
> **b. Motivation of equilibrium**
>
> Unlike ViTs and LLMs which exhibit outliers **only** in activations, extreme values are presented in **both** activations and weights in DiTs. Fortunately, they do not coincide in the same activation and weight channels. Such complementarity property inspired us to seek a balance between activations and weights, where their extreme values can be mitigated. This balance is the essence of **equilibrium**. To realize this, we adopt geometric mean as it is suitable for the multiplicative relationship between activations and weights.
>
> **c. Motivation of inverse ρ**
>
> The motivation of inverse Spearman's ρ stems from the need to balance salient weights with salient activations from different timesteps, which have various degrees of complementarity.
>
> Spearman's ρ measures the rank correlation between two variables (the statistical dependence between their rankings). A high ρ suggests that two variables exhibit higher values simultaneously, which **inversely** reflects the complementarity. In our context, the high ρ implies that the same channel tends to have extreme values in both activations and weights, hindering the salience balancing.
>
> To counteract this, our SSC inversely weights timesteps with high ρ between activation and weight salience, thereby prioritizing timesteps with more significant complementarity. This approach helps distribute the extreme values more evenly across channels and timesteps, improving overall quantization performance.
> ## **Question 2: Post-softmax layer**
> We quantize the post-softmax layer.
> ## **Question 3: Generation setting**
> **a. Clarification on FID-10K**
>
> The origin DiT paper [1] achieves a 2.27 **FID-50K** on ImageNet 256x256.
>
> In our work, the only difference is that we use **FID-10K**, following pioneering studies on diffusion such as [2,3,4]. Specifically, the work on IDDPM by OpenAI [3] indicates that while FID-10K may result in a slight increase, it significantly reduces computational resource demands.
>
> Thus, we use FID-10K to facilitate more experiments for comprehensive evaluation. For fair comparisons, all baseline methods and our PTQ4DiT are evaluated using the same metrics.
>
> **b. Experiment on 50K samples**
>
> We perform an additional experiment generating 50K samples on ImageNet 256x256 as in [1], evaluating W8A8 quantization. The results indicate that PTQ4DiT effectively recovers the performance and closely matches the FP model:
> |Method|FID-50K↓|sFID-50K↓|
> |---|---|---|
> |FP|2.27|4.60|
> |PTQ4DiT|2.31|4.82|
>
> In this setting, generating 50K samples takes around 167 hours on an NVIDIA RTX A6000, which is less feasible for extensive comparisons.
>
> [1] Scalable diffusion models with transformers. CVPR 2023.
>
> [2] Diffusion Models Beat GANs on Image Synthesis. NIPS 2021.
>
> [3] Improved denoising diffusion probabilistic models. ICML 2021.
>
> [4] Post-training Quantization on Diffusion Models. CVPR 2023.

---

### Author Rebuttal · Authors · 2024-08-06

# **General Response by Authors**
We express our gratitude to all the reviewers for dedicating their time and providing valuable comments. They acknowledged that our work is novel (N2VF, USPH), effective for DiT quantization (USPH, jrKh, a8C3) and well-written (USPH, jrKh). However, the reviewers also raised constructive concerns about the method's generality and statistical evidence to support our observations. To further enhance our paper, we added the corresponding experiments with analysis and presented them as follows.

## **General Response 1: Generality of PTQ4DiT**
To verify the generality of PTQ4DiT, we extend our experiment to include PixArt-α [1], an advanced Diffusion Transformer model facilitating text-to-image generation. Consistent with the literature convention [1, 2, 3], we adopt the CLIP score as our metric and perform text-to-image generation on the COCO validation dataset.
|Bit-width|Method|CLIP Score↑|
|---|---|---|
|FP|PixArt-α|31.5305|
|W8A8|PTQ4DiT|31.5368|
|W4A8|PTQ4DiT|31.5077|

The results demonstrate that PTQ4DiT significantly recovers the generation ability and delivers comparable performance to FP PixArt-α, suggesting the general efficacy of our method for quantizing Transformer-based diffusion models.

[1] PixArt-α: Fast Training of Diffusion Transformer for Photorealistic Text-to-Image Synthesis. ICLR 2024.

[2] Clipscore: A reference-free evaluation metric for image captioning. EMNLP 2021.

[3] Q-Diffusion: Quantizing Diffusion Models. ICCV 2023.


## **General Response 2: Experiments on direct evidence**
We analyze the effects of our CSB and SSC on channel salience, providing direct quantitative evidence to support their efficacy.

**Experiment design.** For an in-depth evaluation, we design a statistical experiment assessing the percentage of salient channels before and after CSB and SSC. To accurately identify the salient channels, we adopt a robust statistical outlier detector, Interquartile Range (IQR) [1]. This method identifies data points that lie significantly outside the middle 50% of the distribution as outliers. We perform IQR detection on maximal absolute values of channels to identify the salient channels.

**Experiment result.** We assess the percentage of salient channels detected by IQR. For a comprehensive assessment, we average the results over 100 random samples on ImageNet 256x256 generation and across all DiT layers:

| Model | Salient Activation Channel | Salient Weight Channel |
| --- | --- | --- |
| Original DiT | 5.37% | 6.10% |
| + CSB | 3.53% | 3.41% |
| **+ CSB + SSC (PTQ4DiT)** | **1.25%** | **1.32%** |

The results demonstrate a significant reduction in the percentage of salient channels when applying CSB and SSC, highlighting the efficacy of our proposed method in suppressing salient channels.

[1] Outlier detection: Methods, models, and classification. In ACM Computing Surveys (CSUR), 2020.

***For each reviewer's individual concerns, we would like to address them in the responses separately.***

---

### Author Response · Authors · 2024-08-12
**Summary of Authors' Responses**

Dear Reviewers,

We sincerely appreciate the time and effort you have dedicated to reviewing our paper. We have carefully considered all the points raised and would like to summarize our responses as follows:

**(I) Additional experiments.** We conducted three additional experiments: (1) W8A8 ImageNet 512x512 generation, (2) PTQ4DiT on other DiT models, and (3) evaluation using 50K samples. These experiments further validate our method's effectiveness and generality.

**(II) Novelty and motivation.** We highlighted the unique innovations and contributions of our PTQ4DiT and clarified the motivation of the equilibrium and inverse ρ.

**(III) Direct evidence.** We supplemented two statistical experiments: (1) the Jaccard similarity of salient activation and weight channels, which provides direct evidence of the important property, and (2) the reduction in the percentage of salient channels when applying CSB and SSC, supporting their effectiveness.

**(IV) Methodology and implementation.** We clarified the quantization of Post-softmax layers. Moreover, we justified the use of the geometric mean and discussed the extension of PTQ4DiT to other structures besides the linear layers.

***Thank you again for your valuable feedback. We would like to address any remaining concerns for further enhancement of our work.***

---

### Author Response · Authors · 2024-08-13
**Sincere Thanks for Constructive Review and Support**

Dear Reviewers,

We would like to express our gratitude for your recognition of our work and your constructive feedback. Your insightful suggestions have significantly contributed to refining our research and enhancing its overall impact.

We particularly appreciate the time and effort you invested in the review process. We look forward to any further feedback you may have and remain committed to addressing any additional concerns. Thank you.

Sincerely,

The Authors

---

### Decision · Program_Chairs · 2024-09-25

**Decision:**

Accept (poster)

**Comment:**

All reviewers agree on the quality of this paper, I recommend acceptance